# Single-cell transcriptomes of zebrafish germline reveal progenitor types and feminization by Foxl2l

Chen-wei Hsu[1,2†], Hao Ho[3,4†], Ching-Hsin Yang[2], Yan-wei Wang[2], Ker-Chau Li[3,4]*, Bon-chu Chung[1,2,5,6]*

[1]National Center for Biomodels, National Applied Research Laboratories, Taipei, Taiwan; [2]Institute of Molecular Biology, Academia Sinica, Taipei, Taiwan; [3]Institute of Statistical Science, Academia Sinica, Taipei, Taiwan; [4]Statistics Department, University of California, Los Angeles, Los Angeles, United States; [5]Graduate Institute of Biomedical Sciences, China Medical University, Taichung, Taiwan; [6]Neuroscience and Brain Disease Center, China Medical University, Taichung, Taiwan

**Abstract** Zebrafish is an important organism for genetic studies, but its early germ cell types and the mechanism of sex differentiation have not been fully characterized. Here, we profiled single-cell transcriptomes and charted a developmental trajectory going from germline stem cells, through early, committed, and late progenitors, to premeiotic and meiotic cells. We showed that the transcription factor Foxl2l was expressed in the progenitor, directing progenitor differentiation toward oocytes. CRISPR-Cas9-mediated mutation of *foxl2l* produced 100% male fish with normal fertility. Another single-cell profiling of *foxl2l−/−* germ cells revealed the arrest of germ cell development at the stage of progenitor commitment. Concomitantly, *nanos2* transcript (germline stem cell marker) was elevated together with an increase of *nanos2+* germ cells in *foxl2l* mutants, indicating the acquisition of a novel stem cell state. Thus, we have identified developmental stages of germ cells in juvenile zebrafish and demonstrated that zebrafish Foxl2l drives progenitor germ cells toward feminization and prevents them from expressing *nanos2*.

*For correspondence:
kcli@stat.ucla.edu (K-CL);
mbchung@sinica.edu.tw (B-cC)

†These authors contributed equally to this work

**Competing interest:** The authors declare that no competing interests exist.

## Editor's evaluation

Findings have important significance for understanding the role of the *Foxl2*-family of proteins in the early stages of germ cell maturation. Evidence to support the conclusions are solid and use state-of-the-field single cell transcriptomics. Results will interest biologists investigating mechanisms of reproduction, causes of infertility, and the mechanisms by which gonads embark on becoming a testis or an ovary.

## Introduction

Sex differentiation is an important yet complicated subject. The mammalian sex is determined by the XX/XY system, in which the *SRY* gene on the Y-chromosome plays a pivotal role (**Berta et al., 1990**). Females lack *SRY*. They rely on genes such as *RSPO1*, *FOXL2*, and *WNT4* to promote female differentiation and to antagonize male differentiation (**She and Yang, 2017**). In contrast to mammals, fishes have more complicated schemes of sex differentiation. Wild zebrafish strain Nadia uses the female-dominant ZZ/ZW system, but domesticated laboratory strains have no sex chromosomes (**Wilson et al., 2014**). Furthermore, male and female zebrafish are morphologically indistinguishable at the larval stage, creating an obstacle for the study of fish sex differentiation. Zebrafish oocytes start to

develop at around 21 dpf (*Tong et al., 2010*), while male differentiation appears later at around 30 dpf (*Luzio et al., 2021*; *Uchida et al., 2002*). The sex differentiation mechanism in zebrafish remains unclear.

Germ cell abundance and the existence of oocytes play essential roles in the differentiation of zebrafish gonads. Zebrafish containing reduced numbers of germ cells (*nr0b1* mutants) or no germ cells (*dnd* morphants) develop into males (*Chen et al., 2016*; *Slanchev et al., 2005*). Mutations of genes (*figla*, *sycp3,* or *bmp15*) involved in oocyte development also result in female-to-male sex reversal (*Dranow et al., 2016*; *Pan et al., 2022*; *Qin et al., 2018*). These results indicate that intrinsic signals from germ cells are required for the female fate. Germ cell types during and after sex differentiation stages have been well characterized by scRNAseq (*Liu et al., 2022*; *Wilson et al., 2024*). However, the regulation and subtypes of germ cells before meiosis in zebrafish have not been fully characterized.

One transcription factor that promotes female differentiation is Foxl2l (forkhead box L2-like). Medaka (*Oryzias latipes*) *foxl2l*, named *foxl3* (*Liu et al., 2022*), encodes a germline intrinsic factor that suppresses spermatogenesis and initiates oogenesis in XX gonads (*Kikuchi et al., 2020*). XX medaka fish with a *foxl3* mutation become hermaphrodites with both oocyte and functional sperm in adult XX gonads (*Nishimura et al., 2015*). Zebrafish *foxl2l* is a marker of progenitors. Its disruption leads to 100% male fish (*Liu et al., 2022*), signifying the role of Foxl2l in female development. However, the mechanism of Foxl2l action remains poorly understood.

In this study, we conducted single-cell RNA sequencing (scRNAseq) analysis to dissect the developmental trajectory of germ cells in zebrafish during the critical sex determination stage. In addition to the known germline stem cells and meiotic germ cells, we have further identified three subpopulations of progenitor cells: early progenitor, committed progenitor, and late progenitor. We conducted another scRNAseq of *foxl2l* mutant and proved that Foxl2l controls the development of committed progenitors to late progenitor, which is indispensable for female differentiation. We further show that Foxl2l suppresses stemness-related genes in progenitors.

## Results

### Developmental trajectory of zebrafish germ cells

Germ cells play important roles in zebrafish sex differentiation, but we still have much to learn about their development. We set out to investigate their development during the initial stage of sex differentiation at 26 dpf (*Luzio et al., 2021*; *Uchida et al., 2002*). Fluorescent germ cells were isolated by cell sorting from *piwil1:EGFP* transgenic fish, which labels all stages of germ cells with EGFP (*Leu and Draper, 2010*; *Ye et al., 2019*; *Figure 1—figure supplement 1*). The cells were used for scRNAseq using the 10X Genomics platform (*Figure 1A*). After data preprocessing, we obtained single-cell transcriptomes for clustering analysis. The germ cells were partitioned into 14 clusters and visualized on UMAP (*Figure 1B*). Trajectory analysis identified a linear structure for germ cell development, and pseudotime analysis ordered these clusters as W1 to W14 (*Figure 1C*). Marker genes from each cluster were found, and their expression profiles were displayed in a heatmap, showing high expression of marker genes in neighboring clusters in a successive pattern (*Figure 1—figure supplement 2*). We also showed the expression profiles of the top 37 genes with the highest specificity within each cluster out of a total of 1419 marker genes on UMAP separately (*Figure 1—figure supplement 3*).

Using known markers, we annotated zebrafish germ cells to include three broad categories: germline stem cells (GSCs), progenitors (Prog), and meiotic germ cells. Cells in cluster W1 were identified as GSCs because they express *nanos2* (*Figure 1D*; *Beer and Draper, 2013*; *Cao et al., 2019a*). Clusters W3 to W6 were identified as progenitors because they have high cell-cycle scores (*Figure 1E*), and progenitors are known to divide synchronously forming germline cysts (*Bertho et al., 2021*). Cluster W2 lies between W1 and W3 with reduced *nanos2* expression, indicating they transit from GSC to progenitor (G-P). Cells in clusters W8 to W14 are enriched with genes highly expressed in meiotic oocytes (*sycp1*, *sycp3*, *dmc1*, *buc,* and *zp3e*) (*Gautier et al., 2013*; *Pan et al., 2022*; *Marlow and Mullins, 2008*; *Wassarman and Litscher, 2021*; *Yoshida et al., 1998*). Cluster W7 lies between progenitors and meiotic cells; it has a high S-phase score typical of Pre-leptotene (Pro-Lept) at the premeiotic S-phase (*Figure 1E*).

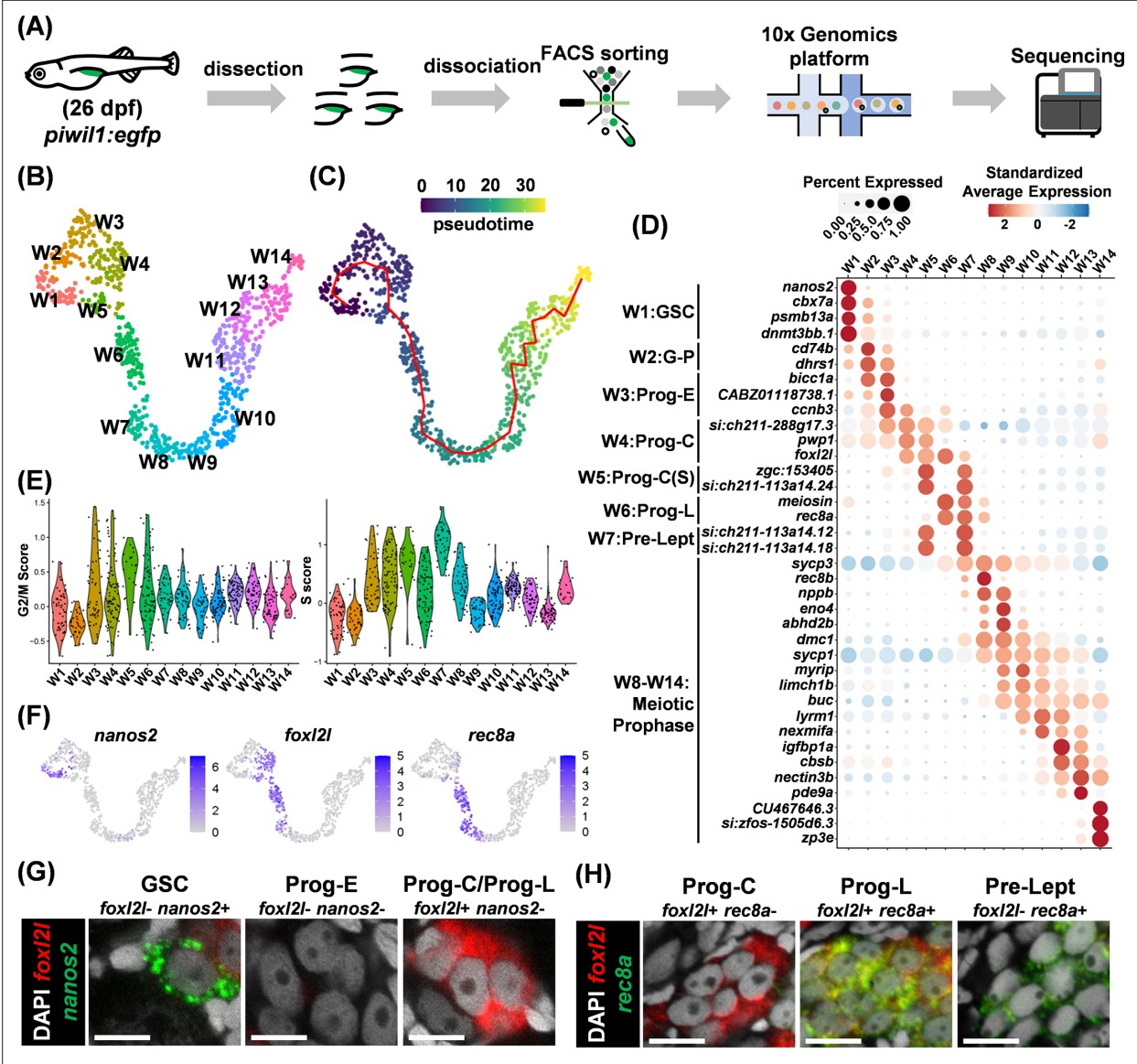

**Figure 1.** Single-cell transcriptome landscape of zebrafish germ cell development. (**A**) Flowchart for sample collection and RNA sequencing. (**B**) *Uniform Manifold Approximation and Projection* (UMAP) visualization of wild-type (WT) germ cells in 14 clusters. (**C**) Pseudotime shown by the color bar and the trajectory shown by a red line. (**D**) Dot plot showing the expression of top marker genes for each cluster. (**E**) Violin plots displaying S and G2/M cell-cycle scores for each cluster. (**F**) UMAP visualization of marker gene expression for *nanos2*, *foxl2l*, and *rec8a*. (**G,H**) Identification of germ cell types by RNA fluorescent in situ hybridization (FISH) of marker genes, *nanos2* and *foxl2l* in (**G**), *rec8a* and *foxl2l* in (**H**), in 26 days post fertilization (dpf) gonads. Scale bar: 10 μm.

The online version of this article includes the following figure supplement(s) for figure 1:

**Figure supplement 1.** Germ cell sorted by fluorescence-activated cell sorting (FACS).

**Figure supplement 2.** Analysis of wild-type (WT) clusters.

**Figure supplement 3.** Expression profiles of top and known marker genes in wild-type (WT) germ cells at 26 days post fertilization (dpf).

**Figure supplement 4.** Expression profiles of top and known wild-type (WT) markers extracted from the database.

A portion of progenitors (W4 to W6) expressed the progenitor gene *foxl2l* (*Figure 1D*), suggesting them as progenitor subtypes. We defined cluster W3 as early progenitors (Prog-E) characterized by the absence of *foxl2l*, cluster W4 as committed progenitors (Prog-C) characterized by the expression of *foxl2l*, cluster W5 as Prog-C(S) characterized by the expression of both *foxl2l* and typical S-phase genes, *zgc:153405* and *si:ch211-113a14.24* (orthologs of linker histone *H1* genes), and cluster W6 as

the late progenitor (Prog-L) characterized by the expression of *foxl2l* and genes for meiotic entry such as *rec8a* (ortholog of *REC8*) and *meiosin* (ortholog of *MEIOSIN*, the meiosis initiator) (**Figure 1D**).

In addition to our own data, we also extracted scRNAseq data of 40 dpf female germ cells from a public database (GSE191137) (**Liu et al., 2022**). Following the same data analysis procedure, the developmental trajectory and marker gene expression profiles were found consistent with ours (**Figure 1—figure supplement 4**), suggesting the validity of our data and the finer developmental transitions of germ cells in juvenile gonads as charted above.

## Histological identification of germ cell progenitors

Zebrafish progenitors are round in shape with one to three nucleoli at the center of the nucleus, and morphologically indistinguishable from GSC (**Tong et al., 2010**). Our single-cell transcriptome analysis identified markers *nanos2*, *foxl2l*, and *rec8a*, which mark different subtypes of germ cells (**Figure 1F**). This can be verified by double fluorescence in situ hybridization (FISH). GSC was only positive for *nanos2* (*foxl2l*⁻*nanos2*⁺). Prog-E was negative for both *nanos2* and *foxl2l* (*foxl2l*⁻*nanos2*⁻), whereas both Prog-C and Prog-L were *foxl2l*⁺*nanos2*⁻ (**Figure 1G**). Double FISH also detected Prog-C as *foxl2l*⁺ *rec8a*⁻, Prog-L as *foxl2l*⁺ *rec8a*⁺, and pre-leptotene cells as *foxl2l*⁻ *rec8a*⁺ (**Figure 1H**). The FISH result is consistent with the scRNAseq data, which showed little overlap between *foxl2l* and *nanos2* expression in different types of germ cells (**Figure 1—figure supplement 3**).

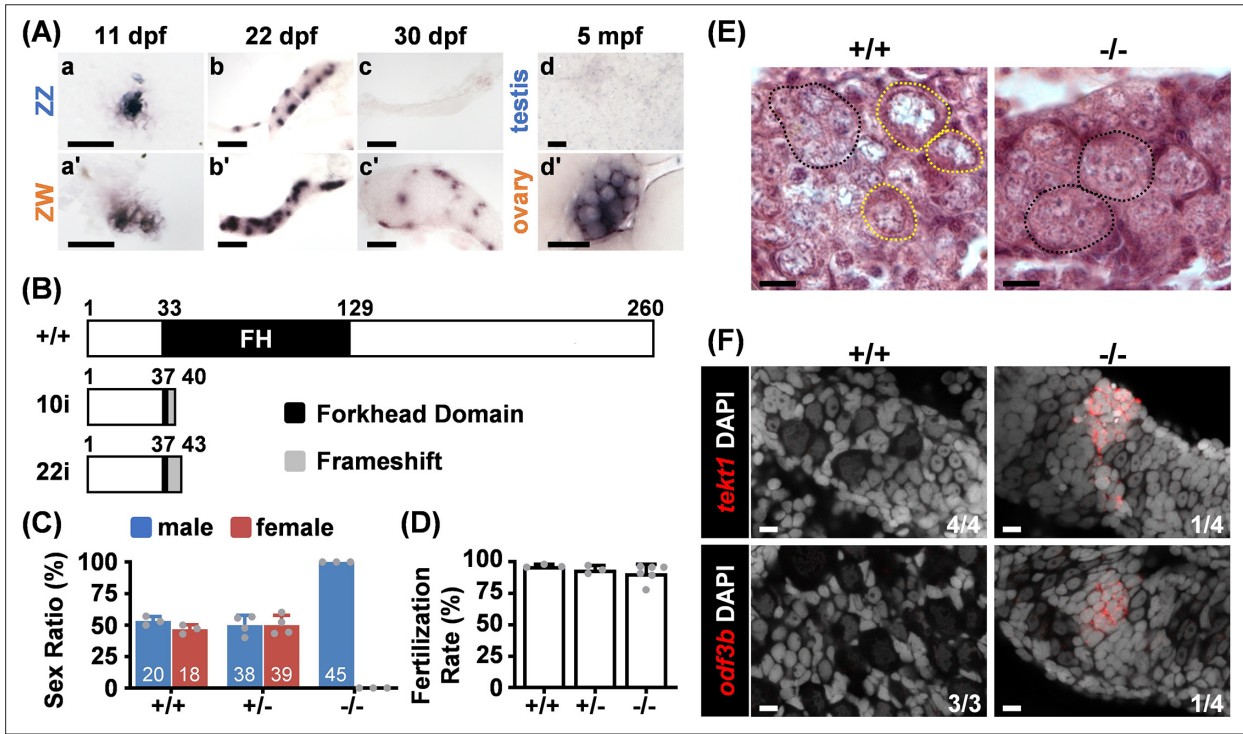

**Figure 2.** Foxl2l is required for female differentiation. (**A**) In situ hybridization detects *foxl2l* transcripts in indifferent gonads at 11 days post fertilization (dpf) (**a, a'**). Expression of *foxl2l* becomes female biased when female differentiation initiates at 22 dpf (**b, b'**), and becomes female-specific at the initiation of male differentiation at 30 dpf (**c, c'**). In adults, *foxl2l* is expressed in cystic cells in ovary but not in testis (**d, d'**). mpf: months post fertilization. All samples were collected from Nadia strain. Scale bars are 50 µm in (**a, a'**), 100 µm in (**b, b', c, c'**), 20 µm in (**d, d'**). (**B**) Domain structure of wild-type (WT) and the predicted mutant of Foxl2l with 10 base pairs insertion (10i) or 22 base pairs insertion (22i) generated by CRISPR/Cas9 system. (**C**) Sex ratios of WT (+/+), *foxl2l*⁺/²²ⁱ heterozygous (+/-), and *foxl2l*²²ⁱ/²²ⁱ homozygous (-/-) mutants at 4 months of age reveal that *foxl2l* mutant fish are all males. The total numbers of fish in each genotype and gender are labeled in white. Dot: sex ratio of each batch. (**D**) All *foxl2l*¹⁰ⁱ/¹⁰ⁱ mutant males (-/-) are fertile. Dot: individual fish used in the test, n=3 in WT and heterozygous mutants, n=6 in homozygous mutants. (**E**) The *foxl2l*²²ⁱ/²²ⁱ homozygous (-/-) mutants lack meiotic oocytes at 20 dpf shown by histological staining. Black dashed circle: germ cell cysts with one to three nucleoli in each cell. Yellow dashed circle: meiotic oocyte. Scale bars represent 20 µm. (**F**) The *foxl2l*²²ⁱ/²²ⁱ mutants (-/-) express male markers (*tekt1* or *odf3b*) detected by RNA fluorescence in situ hybridization (FISH) at 28 dpf. Scale bars are 10 µm.

The online version of this article includes the following figure supplement(s) for figure 2:

**Figure supplement 1.** Generation and the phenotype analysis of *foxl2l* mutant.

## Expression of *foxl2l* becomes female biased during sexual differentiation

We examined the temporal expression of *foxl2l* by in situ hybridization during the sexual differentiation period. We used the Nadia strain for its defined ZZ/ZW (male/female) sex chromosomes, facilitating the distinction of their sexes at the juvenile stage (*Wilson et al., 2014*). The *foxl2l* expression was similar between ZZ and ZW gonads at 11 dpf before apparent sex differentiation (*Figure 2A*). Upon female differentiation at 22 dpf, *foxl2l* expression was decreased in the male ZZ gonad. The female-biased expression became more obvious at 30 dpf. In the adult ovary at 5 months post fertilization, *foxl2l* was highly expressed in cystic germ cells but was not observed in testis (*Figure 2A*).

## All *foxl2l* mutants fail to enter female meiosis and become fertile males

To investigate the function of *foxl2l*, we generated *foxl2l* mutants using CRISPR-Cas9. Two resultant mutant lines, *foxl2l*$^{10i/10i}$ and *foxl2l*$^{22i/22i}$, contained DNA insertions (*Figure 2—figure supplement 1*), causing a premature stop codon disrupting the forkhead (FH) domain of Foxl2l (*Figure 2B*). All *foxl2l* mutants became males (*Figure 2C*), and all *foxl2l* mutant males were fertile (*Figure 2D*). We previously also showed that the mutant and wild-type (WT) testes are similar in histology (*Yang et al., 2025*).

We further examined the histology of mutant gonads. At 20 dpf, diplotene oocytes with large perinucleolar nuclei were observed in WT gonads, but only non-meiotic cystic germ cells were found in *foxl2l* mutants (*Figure 2E*). At 28 dpf, small and round germ cells that express spermatogenic markers, *tekt1* and *odf3b* (*Nishimura et al., 2015*), were found in mutant but not WT gonads (*Figure 2F*). Therefore, in *foxl2l* mutants, germ cells fail to enter female meiosis and develop into fertile males. This indicates that *foxl2l* is required for female but not for male development.

## The development of *foxl2l* mutant germ cells is halted at the committed progenitor stage

We conducted scRNAseq from 26 dpf *foxl2l*$^{-/-}$ germ cells to investigate their developmental stages via transcriptomic profiling. The expression profiles of the top 37 WT marker genes in mutant cells were visualized via UMAP (*Figure 3—figure supplement 1A*). Marker genes corresponding to WT clusters of GSC, G-P, Prog-E, and Prog-C were upregulated in distinct subsets of mutant cells, whereas markers associated with stages beyond Prog-L were expressed at consistently low levels. Accordingly, we inferred the developmental stages of mutant cells by aligning their transcriptomic profiles with those of WT cells (*Figure 3—figure supplement 1B, C*).

Pie charts showed that most of the mutant cells were annotated as GSC, G-P, Prog-E, or Prog-C, with only a few as Prog-C(S), the stage preceding Prog-L differentiation (*Figure 3A*). None of the mutant cells were at Prog-L or meiotic stages (*Figure 3A*). This indicates that mutant germ cells cannot progress beyond Prog-C. To verify this statement, we used fluorescent staining and detected Prog-L marker *rec8a* in cystic germ cells and meiotic marker Sycp3 in meiotic cells of the control (*Figure 3B*). Both *rec8a* and Sycp3 staining were missing in *foxl2l* mutant gonads. These data show that mutant germ cells do not enter the state of Prog-L. The components of WT cells from GSC to Prog-C stages were examined more closely (*Figure 3—figure supplement 2A*). The data of these cells were used for further analysis with mutant germ cells.

## Altered developmental program in *foxl2l* mutant

The mutant germ cells exhibited heterogeneity, encompassing diverse cell types. To better understand mutant subtypes and their relationship with WT cells, we conducted a co-clustering analysis of the full transcriptomic profiles of both WT cells (GSC to Prog-C(S) stages) and mutant cells, as outlined in *Figure 3—figure supplement 2B*. This approach does not assume that mutant cells correspond directly to specific WT stages; instead, it enables the identification of potentially novel germ cell types. This analysis also provides insight into the relationships between WT and mutant cells, resulting in eight integrated groups (I1-I8) (*Figure 3C*). WT and mutant cells with similar transcriptomic profiles were grouped together within the same integrated cluster.

We examined the distribution of cells at different stages across different integrated groups (*Figure 3D*). Prominent stages were identified after calibrating against their prevalence rates within the entire WT or mutant cell population. For WT cells, we found WT GSC is prominent in groups I1

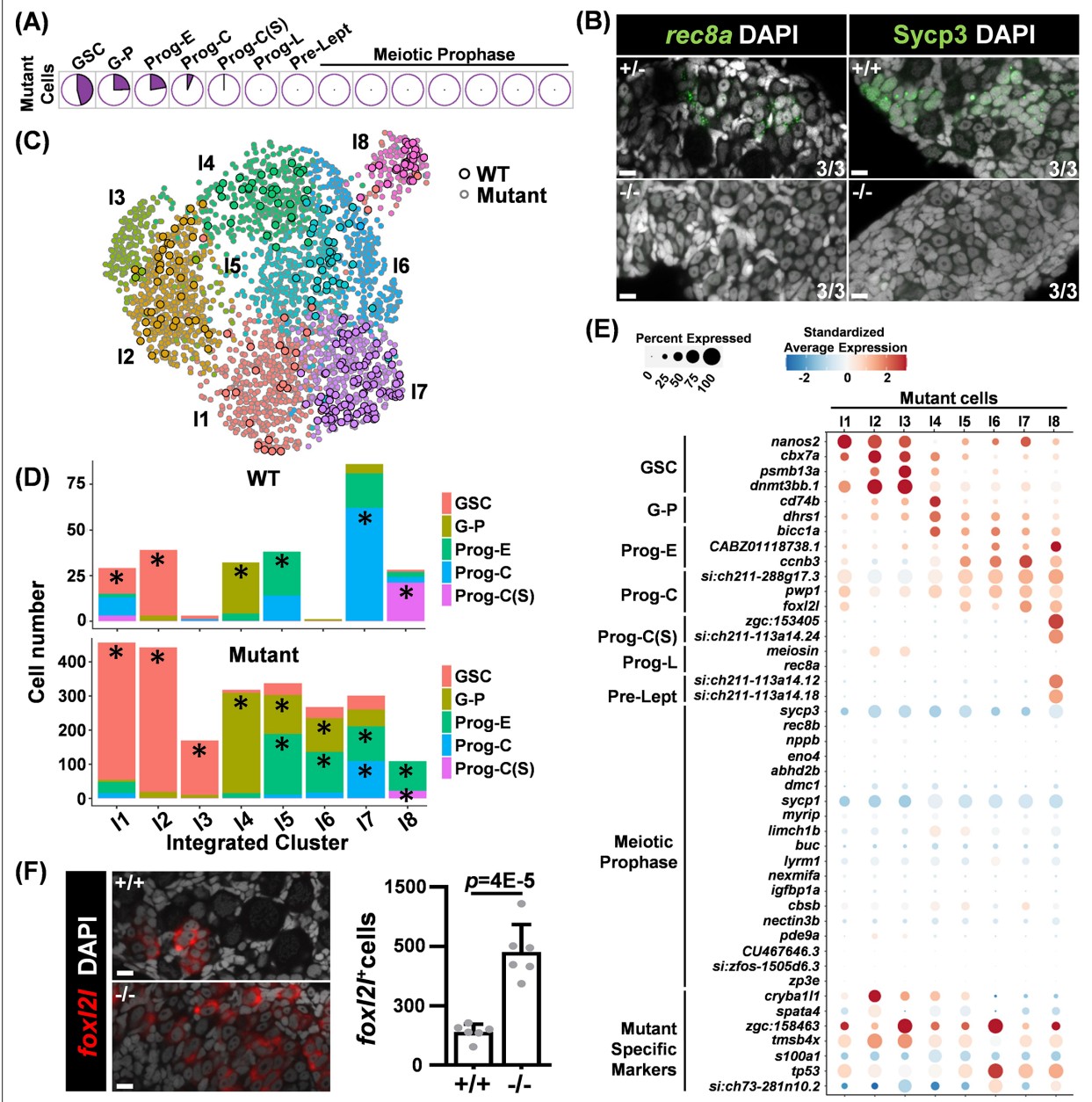

**Figure 3.** Germ cells in *foxl2l* mutants are arrested at the stage of committed progenitor with aberrant gene expression. (**A**) Pie charts showing the proportion of mutant cells classified into different developmental stages. (**B**) Absence of the Prog-L marker *rec8a* and meiotic marker Sycp3 in *foxl2l* mutant gonads. (Left) RNA fluorescence in situ hybridization (FISH) and (right) immunofluorescence staining of wild-type (WT) (+/+), *foxl2l*+/10i (+/-), and *foxl2l*10i/10i (-/-) gonads at 28 days post fertilization (dpf) (left) and 21 dpf (right), respectively. The numbers at the bottom right corner indicate that all three gonads in each genotype have the same staining pattern. Scale bars represent 10 μm. (**C**) Uniform Manifold Approximation and Projection (UMAP) visualization of integrated cells grouped into eight clusters. Circles with black outline: WT cell. Circles with gray outline: mutant cell. (**D**) Bar graphs showing the proportions of developmental stages for WT and mutant cells in each integrated group. Prominent stages with statistical significance were marked by asterisks (*). (**E**) Dot plot showing the expression of marker genes (Y-axis) in the mutant cells across integrated groups (X-axis). (**F**) *foxl2l*10i/10i (-/-) gonads contain an increased number of *foxl2l*+ cells at 26 dpf. Left: in situ hybridization of the gonads with *foxl2l*. Right: quantification of the staining data. One dot in the graph represents one gonad, n=6 in each genotype.

The online version of this article includes the following figure supplement(s) for figure 3:

**Figure supplement 1.** Transcriptome analysis of mutant germ cells.

**Figure supplement 2.** Transcriptome analysis of integrated groups.

*Figure 3 continued on next page*

**Figure supplement 3.** The flowchart of data processing and cell-cycle scoring for wild-type (WT) and mutant single-cell RNA sequencing (scRNAseq) transcriptome.

**Figure supplement 4.** The flowchart of cell clustering analysis for wild-type (WT) cells.

and I2, G-P is prominent in group I4, Prog-E is prominent in group I5, Prog-C is prominent in group I7, and Prog-C(S) is prominent in group I8 (*Figure 3D*).

For mutant cells, groups I5-I8 had two prominent stages that are similar to those in WT (*Figure 3D*). This indicates that WT and mutant cells that have high transcriptome similarity likely belong to the same developmental stage. Groups I1 and I2 were dominated by GSC, but the expression patterns of top GSC marker genes were slightly different with higher expression of *nanos2* in I1 but of *cbx7a*, *psmb13a,* and *dnmt3bb.1* in I2 (*Figure 3—figure supplement 2C*). This suggests that GSCs can be further divided into two finer stages, GSCI and GSCII.

Groups I3 and I6 are different from the rest of the groups. They contained few WT cells and were classified as GSC for I3 and as Prog-E for I6 mutant cells (*Figure 3D*). In addition to expressing the GSC marker, I3 mutant cells expressed *zgc:158463*; I6 mutant cells expressed *tp53* and *zgc:158463* in addition to Prog-E markers (*Figure 3E*). This indicates that I3 and I6 mutant cells expressed different genes from WT GSCs and progenitors, potentially leading to a unique cell fate.

Furthermore, the Prog-C marker *foxl2l* was expressed in mutant cells of I1, I5, I7, and I8, which were mostly classified as GSC or Prog-E in addition to Prog-C (*Figure 4E*). The number of *foxl2l+* cells detected by FISH was also increased in mutant gonads compared to WT (*Figure 3*). This result indicates that mutant *foxl2l* mRNA remains stable. The aberrant *foxl2l* expression in some earlier GSC and Prog-E, the accumulation of *foxl2l*-expressing cells in the mutant, and the failure to find clusters representing more mature stages indicate that progenitor development is impaired in *foxl2l* mutants.

## Differential gene expression between WT and *foxl2l* mutant germ cells

We further evaluated the effect of *foxl2l* mutation on gene expression throughout development. We compared WT and mutant cells at the same prominent stages within each integrated group, which is defined as a stage-matched set, color-coded GSCI, GSCII, G-P, Prog-E, Prog-C, and Prog-C(S) (*Figure 4A*). *Figure 4B* shows all the differentially expressed genes (DEGs) of mutants relative to WT (mutant/WT). Prog-C(S) contained the largest number of DEGs, consistent with the failure of mutant Prog-C(S) cells to differentiate further to Prog-L (*Figure 4B*). We also unexpectedly detected *foxl2l* transcripts in mutant Prog-E (*Figure 4C*). Concomitantly, the expression of *stk31* was decreased and that of *ptmab* was increased upon transition to Prog-C in the WT cells; yet these genes were steadily expressed in the mutant progenitors (*Figure 4—figure supplement 1*). This pinpoints the defect in *foxl2l* mutant germ cells during the transition from Prog-E to Prog-C.

The top DEGs, *dmrt1*, *dele1,* and *id1*, and the GSC marker *nanos2* were further analyzed (*Figure 4C*). The *id1* and *nanos2* transcripts were low in the WT progenitors but became abundant in mutant Prog-C (*Figure 4C*). The expression of *dmrt1* and *dele1* in WT was steady throughout the developmental stages (*Figure 4C*), but *dmrt1* was abnormally high, and *dele1* was abnormally low in both mutant GSCI and Prog-C/Prog-C(S). These results indicate multiple defects of gene expression in the *foxl2l* mutants and suggest diverse roles of Foxl2l, as an inhibitor for *dmrt1* but activator for *dele1*.

## Co-expression of *foxl2l* with *dmrt1*, *nanos2*, or *id1* in *foxl2l* mutants

The expression patterns of *foxl2l* along with *nanos2*, *id1*, and *dmrt1* at the early stages were further examined. The *nanos2* and *foxl2l* genes were distinctly expressed among WT cells, but co-expressed in some mutant cells (*Figure 5A*). Double FISH also detected cells expressing both *nanos2* and *foxl2l*, and the proportion of *nanos2+foxl2l+* cells among *foxl2l+* cells was increased in the mutant (*Figure 5B*). Moreover, the number of *nanos2+* cells was increased in mutants at 21 dpf during the early stage of female differentiation (*Figure 5C*).

The co-expression of the GSC marker, *nanos2*, in the *foxl2l*-expressing Prog-C in mutants raised the hypothesis that GSCs may continue the GSC program during the differentiation into progenitors. To validate this hypothesis, we examined the expression profiles of the GSC marker genes, *nanos2*, in all mutant cells and found that *nanos2* expression was low in most mutant cells classified as G-P and Prog-E (mutant cells of I4 to I6 and I8) (*Figure 3E*). Other GSC markers including *cbx7a*, *psmb13a*,

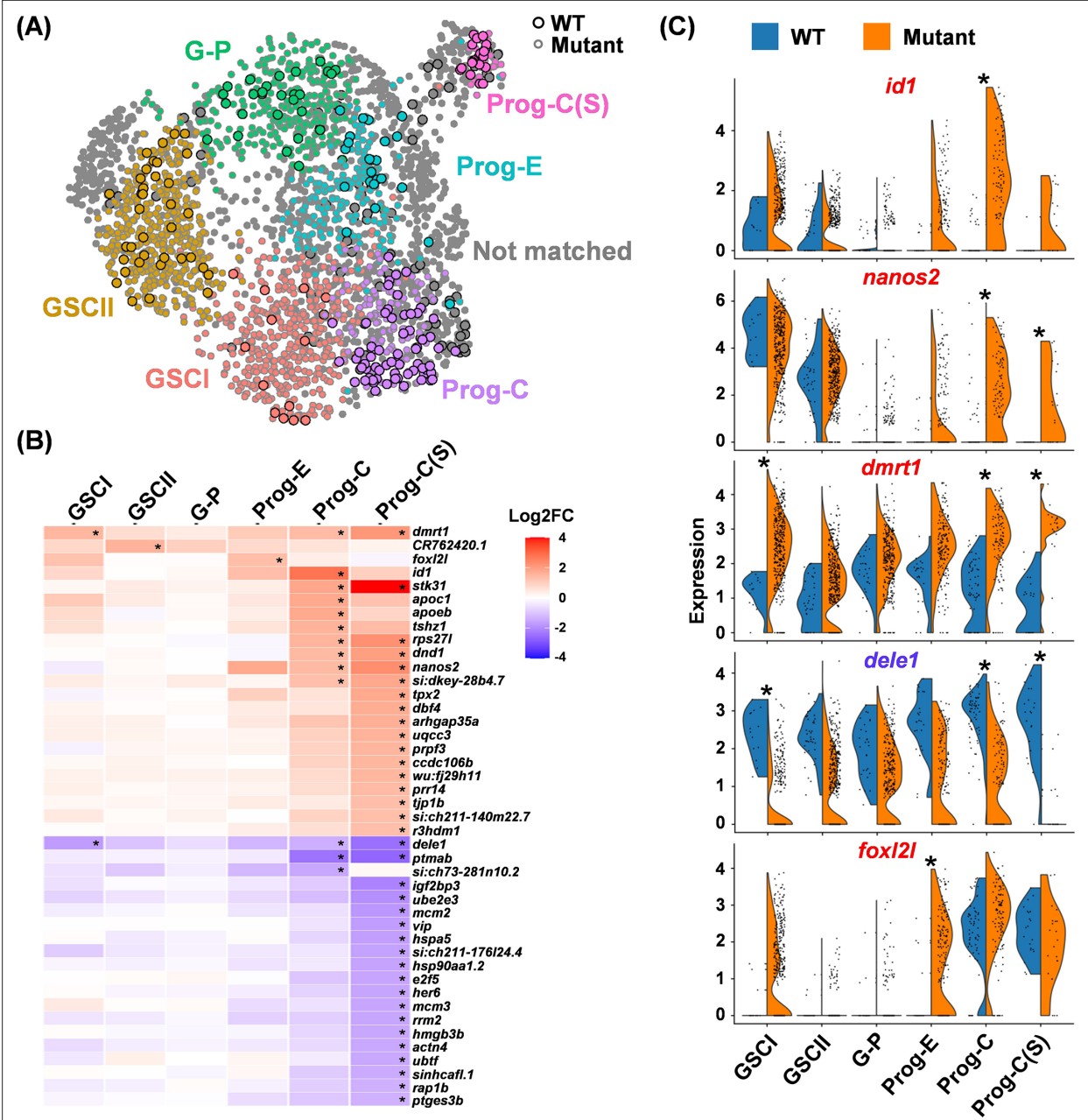

**Figure 4.** Identification of differentially expressed genes (DEGs) in wild-type (WT) and *foxl2l* mutants. (**A**) Uniform Manifold Approximation and Projection (UMAP) visualization of integrated cells, with six matched stages of WT and mutant cells displayed in different colors. Unmatched cells are labeled in gray. Circles with a black outline: WT cells. Circles with gray outline: mutant cells. (**B**) Heatmap showing the fold change (FC) between the mutant versus WT (mutant/WT) for all DEGs at each stage. Developmental stages are displayed on the X-axis, and gene names are displayed on the Y-axis. (**C**) Split-violin plots showing the distribution of cells expressing top DEGs at different development stages (X-axis) between WT and mutant. Asterisks in (**B**) and (**C**) indicate significant differences in gene expression between WT and mutants.

The online version of this article includes the following figure supplement(s) for figure 4:

**Figure supplement 1.** Expression of differentially expressed genes (DEGs) in wild-type (WT) and mutant germ cells.

and *dnmt3bb.1* also showed low expression in those mutant cells mostly classified as different types of progenitors (mutant cells of I4 to I8) (*Figure 3E*). This result indicates that the GSC program in mutant cells does not continue when GSCs are differentiated into progenitors; instead, some *foxl2l*-expressing Prog-C may acquire some GSC properties by upregulating *nanos2* in mutants.

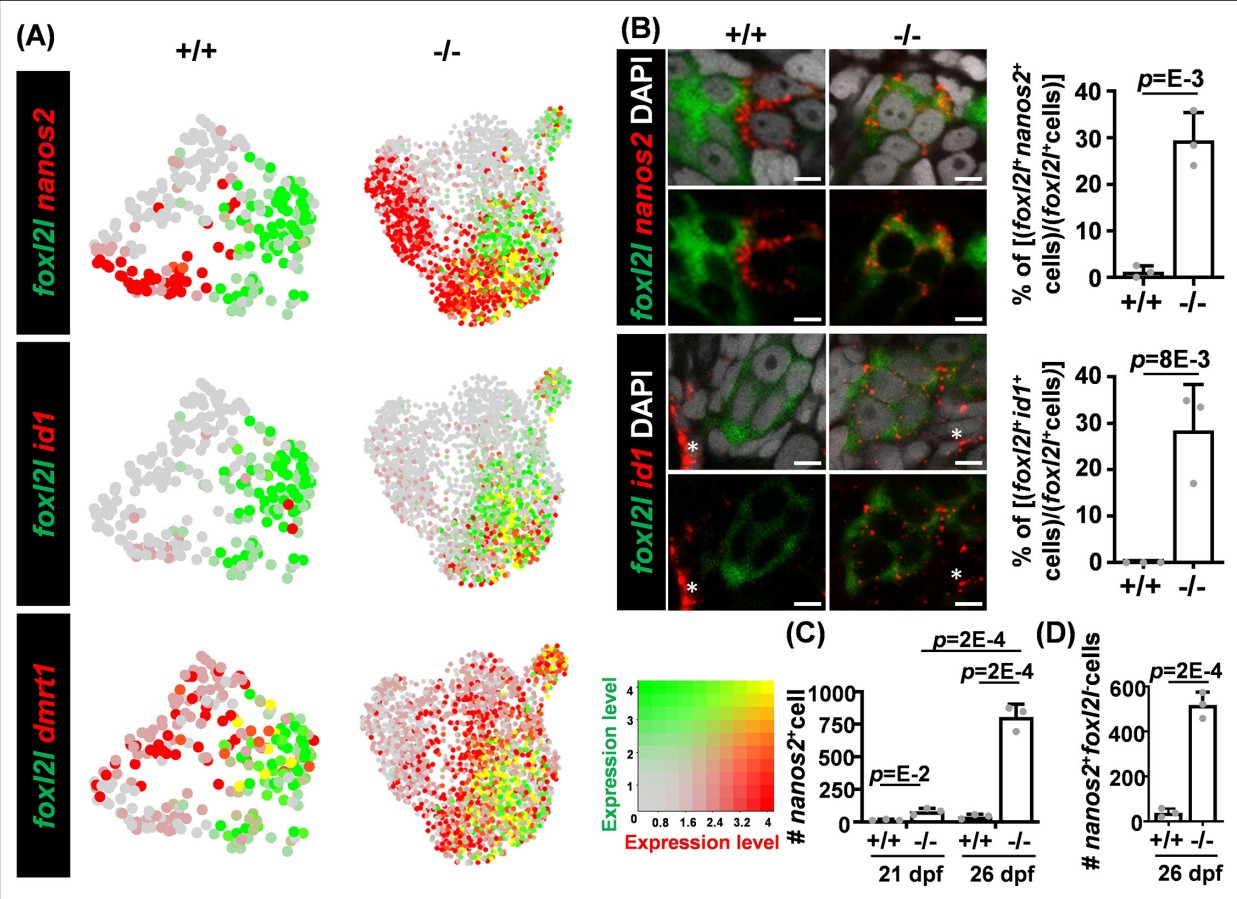

**Figure 5.** Aberrant co-expression of *foxl2l* with *dmrt1*, *id1,* and *nanos2*. (**A**) Uniform Manifold Approximation and Projection (UMAP) visualization of *dmrt1*⁺, *nanos2*⁺, or *id1*⁺ cells and *foxl2l*⁺ cells. Left: wild-type (WT) cells in clusters W1 to W5 (+/+). Right: *foxl2l* mutant (-/-) cells. Yellow dots indicate cells co-expressing both genes. (**B**) Increased proportions of *foxl2l* mutant cells co-expressing *foxl2l* with *nanos2* or *id1*. (Left) Double RNA fluorescence in situ hybridization (FISH) detecting *nanos2* or *id1* with *foxl2l* in WT (+/+) or *foxl2l*^10i/10i^ homozygous mutant (-/-) gonads at 26 days post fertilization (dpf). Stained images are shown with (top panels) or without (bottom panels) DAPI in each set. Asterisks represent the expression of *id1* in somatic cells. Scale bars represent 5 μm. (Right) Quantitation of the proportion of double positive cells in each gonad is shown. One dot represents the data from one gonad, n=3 in each genotype of each graph. (**C**) Increased numbers of *nanos2*-expressing cells in *foxl2l*^10i/10i^ homozygous mutant gonad (-/-). One dot represents the data from one gonad, n=3 in each genotype of each age. (**D**) Increased number of *nanos2*⁺*foxl2l*⁻ germline stem cells (GSCs) in *foxl2l*^10i/10i^ homozygous mutant gonad (-/-). One dot represents the data from one gonad, n=3 in each genotype.

The online version of this article includes the following figure supplement(s) for figure 5:

**Figure supplement 1.** Very few germ cells express *foxl2l* together with *nanos2*, *id1*, or *dmrt1* in 40 days post fertilization (dpf) wild-type (WT) gonad.

To further investigate whether the upregulation of GSC marker *nanos2* in mutant Prog-C alters cell property, we counted the number of *nanos2*⁺*foxl2l*⁻ GSC cells and found it was increased in mutants at 26 dpf compared with WT (***Figure 5D***). Since *foxl2l* is not expressed in WT GSCs, the increased GSC number in *foxl2l* mutants may be due to either direct reversion from Prog-C to GSCs or the proliferation of GSCs triggered by mutant Prog-C. Another gene, *id1*, was barely expressed in WT germ cells but was abundant in some *foxl2l*⁺mutant germ cells (***Figure 5A***). Double FISH further detected an increased proportion of *id1*⁺*foxl2l*⁺ cells in the mutant (***Figure 5B***). Id1 is involved in the maintenance of stemness in various organs (***Kantzer et al., 2022***; ***Ying et al., 2003***; ***Zhang et al., 2014***), consistent with the hypothesis that mutant Prog-C reverts to the GSC property.

Dmrt1 is a male regulator in many vertebrate species (***Ge et al., 2017***; ***Matson et al., 2011***; ***Smith et al., 2009***). In WT, very few cells co-expressed *foxl2l* and *dmrt1*. However, many mutant cells expressed both *foxl2l* and *dmrt1* (***Figure 5A***). Similar results can also be observed in scRNAseq data extracted from a public database (GSE191137), which showed very few 40 dpf female germ cells co-expressed *foxl2l* with *nanos2*, *id1*, or *dmrt1* (***Figure 5—figure supplement 1***).

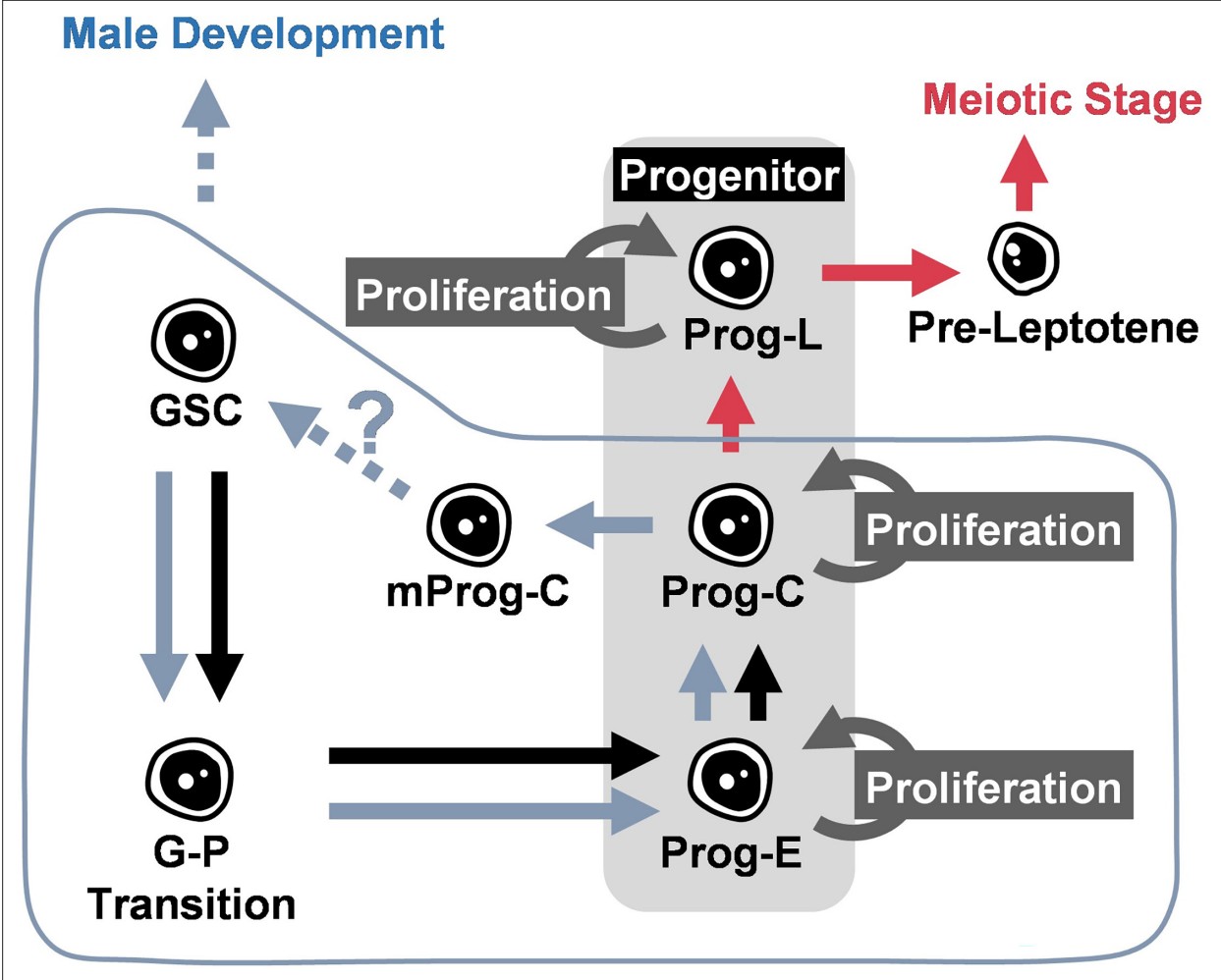

**Figure 6.** Schematic illustration of the function of Foxl2l during germ cell development in juvenile gonad. During the sex determination period, germ cells develop following the trajectory from germline stem cell (GSC), GSC to progenitor (**G–P**) transition, early progenitor (Prog-E), committed progenitor (Prog-C), late progenitor (Prog-L) to the female meiotic stage. Foxl2l is essential for the maturation of Prog-C and the ensuing Prog-L development. In *foxl2l* mutant, germ cells are arrested at Prog-C. Some mutant Prog-C cells (mProg-C) express *nanos2*. The absence of functional Foxl2l eventually triggers male development. Blue symbols indicate alternative development in *foxl2l* mutants. Red arrows indicate the developmental trajectory of females.

## Discussion

In this article, we have analyzed the development of zebrafish germ cells and elucidated the mechanism controlling sex differentiation in zebrafish by analyzing *foxl2l* mutants. The loss of functional Foxl2l leads to gene dysregulation, a halt of Prog-C development, aberrant upregulation of GSC marker gene *nanos2*, and eventual male development. These results show that Foxl2l is required for the differentiation of Prog-C to Prog-L, and therefore guides the progenitor toward the female pathway. Foxl2l probably also prevents progenitors from reverting to the stem cell fate as depicted in *Figure 6*.

### Developmental trajectory of germ cells

The process of zebrafish early germ cell development during the sex differentiation period is poorly understood. Here with scRNAseq analysis, we have depicted the developmental trajectory of germ cells and further characterized GSCs and progenitors. GSCs can be split into two subtypes: GSC-I ($nanos2^{hi}cbx7a^{lo}$) and GSC-II ($nanos2^{lo}cbx7a^{hi}$). GSCs can self-renew intermittently through two types of cell division to ensure continuous gamete maintenance (*Saito et al., 2007*). One type of cell division is symmetric, producing two identical stem cells. The other type is asymmetric division producing

two different daughter cells, one stem cell and the other cell preparing for further differentiation. The two subtypes of GSCs may represent two different daughter cells derived from asymmetric division. This speculation needs to be validated by further analysis. When GSCs lose stem cell property, they differentiate into progenitors.

Progenitors include three types: Prog-E (*foxl2l⁻nanos2⁻*), Prog-C (*foxl2l⁺rec8a⁻*), and Prog-L (*foxl2l⁺rec8a⁺*). We show here that Prog-L is essential for female differentiation. Prog-C is the gate before female differentiation and will stay at S-phase temporarily before developing into Prog-L. In Prog-L, genes such as *rec8a* and *meiosin* are highly expressed for the preparation of meiotic entry. Our study delineates the developmental process of zebrafish germ cells during sex differentiation.

## A type of zebrafish germ cell progenitor possesses female identity at juvenile age

In many vertebrate species, sex determination is induced by sex-determining genes expressed in supporting cells (*Berta et al., 1990*; *Matsuda et al., 2002*). The fate of germ cells is determined following the instruction of supporting cells. In mammals, pre-Sertoli cells express the male sex-determining gene *Sry* to activate a cascade of male-related genes, including *Sox9* and *Amh*, and to trigger male fate decision (*She and Yang, 2017*). In female supporting cell precursors, the absence of *Sry* expression enables the expression of female genes and female fate determination. In contrast, germ cell abundance is required for female development in zebrafish, and the early depletion of germ cells at the embryonic stage results in male development (*Campbell et al., 2015*; *Dai et al., 2017*; *Draper et al., 2007*; *Slanchev et al., 2005*; *Zhou et al., 2018*). Our study further determines early establishment of zebrafish germline feminization in Prog-L. Foxl2l drives the differentiation of Prog-C to Prog-L and, therefore, dictates female development in juvenile gonads.

Medaka Foxl2l controls genes in meiosis and folliculogenesis (*Kikuchi et al., 2020*). Zebrafish Foxl2l may have the same functions as shown by our previous RNAseq result (*Yang et al., 2025*) and the expression of *foxl2l* in Prog-L (**Figure 1**). However, there are differences regarding the mutant phenotypes in zebrafish and medaka. Medaka *foxl2l* (termed *foxl3*) mutant gonads still contain some meiotic oocytes (*Nishimura et al., 2015*). Zebrafish *foxl2l* deficiency, however, leads to germ cell arrest at the Prog-C stage as shown here (**Figure 3A**). No meiotic oocytes can be detected in zebrafish *foxl2l⁻/⁻* gonads. The difference in *foxl2l* mutant phenotypes between zebrafish and medaka raises two possibilities. First, zebrafish Foxl2l has an additional role in driving progenitor differentiation. Second, medaka Foxl2l plays a similar role in progenitor differentiation as in zebrafish; however, medaka may have a stronger female genetic tendency that prevents full blockage of female differentiation and meiosis in *foxl3⁻/⁻* gonads.

## Direct male differentiation from indifferent gonads

Zebrafish testis development has been an unsolved issue. Zebrafish testis can be derived from ovary-like tissue that undergoes oocyte apoptosis during sex differentiation (*Uchida et al., 2002*). Here, we show that germ cells are arrested in the middle of progenitor development followed by direct male differentiation in *foxl2l* mutant. This indicates that zebrafish testis can be differentiated directly from indifferent gonads without going through the female phase. Our current result is also consistent with the reports showing direct male differentiation in zebrafish (*Pan et al., 2022*; *Tong et al., 2010*). The wild zebrafish strain, Nadia, with ZZ sex chromosomes undergoes direct male development without going through female-to-male transition (*Wilson et al., 2024*). Therefore, in addition to female-to-male transition, zebrafish males can develop directly similar to most gonocharists whose males and females are derived directly from undifferentiated gonads.

## Foxl2l suppresses genes in male development and stemness

Our scRNAseq analysis has identified *nanos2*, *dmrt1,* and *id1*, which may be involved in cell-type transition in *foxl2l* mutants. Nanos2 functions in germ cell development in various species (*Cao et al., 2019a*; *Kusz et al., 2009*; *Tsuda et al., 2003*). NANOS2 also suppresses meiosis in male mice (*Suzuki and Saga, 2008*). Therefore, the aberrantly induced *nanos2* in Prog-C of *foxl2l* mutant may participate in the suppression of meiotic genes in addition to the induction of stemness.

Id1, inhibitor of DNA binding 1, is a transcription regulator that lacks a DNA binding domain. It can bind basic helix-loop-helix proteins to prevent their binding to DNA (*Benezra et al., 1990*). Id1 is

essential for homeostasis and the maintenance of stem cells in several cell types (*Hong et al., 2011*; *Kantzer et al., 2022*; *Ying et al., 2003*; *Zhang et al., 2014*). Although the function of Id1 in the testis remains unclear, its paralog Id4 regulates the self-renewal of spermatogonial stem cells in the testis (*Sablitzky et al., 1998*; *Wang et al., 2018*). Thus, the enhanced *id1* expression in the mutant Prog-C germ cells may inhibit genes controlling differentiation and guide cells toward stem cell fate in the *foxl2l* mutant.

Dmrt1 regulates male differentiation in many animals. In Nile tilapia (*Oreochromis niloticus*), Dmrt1 and Foxl2l antagonize each other to determine germline sex (*Dai et al., 2021*). Zebrafish appear to behave similarly. As a male-promoting gene (*Webster et al., 2017*), zebrafish *dmrt1* is aberrantly upregulated in *foxl2l* mutants at the Prog-C, Prog-C(S), and GSCI stage (*Figure 4C*). This aberrant *dmrt1* expression may direct *foxl2l* mutants toward male development.

We show here that Foxl2l regulates genes important for progenitor differentiation from Prog-C to Prog-L (*Figure 4B*); it also suppresses GSC gene *nanos2* and antagonizes male development by suppressing *dmrt1*. In *foxl2l* mutants, most of the DEGs were detected in Prog-C/Prog-C(S), consistent with the failure of the Prog-C differentiation to Prog-L. Foxl2l facilitates the expression of oocyte genes to ensure female differentiation.

Our DEG analysis identified both upregulated and downregulated genes in *foxl2l* mutants (*Figure 4B*), suggesting that Foxl2l can activate or suppress transcription. The dual role of Foxl2l is comparable with that of other Fox family proteins. The forkhead transcription factor L2 (forkhead box L2, FOXL2) is highly expressed in granulosa cells required for female differentiation, including meiosis and folliculogenesis (*Georges et al., 2014*). In mammalian granulosa cells, Foxl2l activates *Fst* to facilitate folliculogenesis (*Kashimada et al., 2011*). However, in undifferentiated pre-granulosa cells, FOXL2 suppresses granulosa differentiation and proliferation by repressing *StAR*, *CYP11A*, *CYP19*, and *CCD2* (*Kuo et al., 2011*; *Kuo et al., 2012*; *Pisarska et al., 2004*; *Pisarska et al., 2010*). In juvenile zebrafish gonads, the activation and suppression of genes involved in progenitor development by Foxl2l results in eventual gonad differentiation.

## Altered development of *foxl2l* mutant germ cells

We have detected aberrantly enriched *tp53* and *zgc:158463* in distinct *foxl2l* mutant cells in I3 or I6. Tumor suppressor protein p53, encoded by *TP53*, induces cell-cycle arrest and apoptosis. Mutation of genes such as *zar1*, *fancl,* and *brca2* causing apoptosis of either early germ cells or developing oocytes results in eventual male development. This mutant phenotype can be rescued by additional *tp53* mutation in these mutants (*Rodríguez-Marí et al., 2010*; *Rodríguez-Marí et al., 2011*; *Shive et al., 2010*). However, additional mutation of *tp53* fails to rescue the sexual-reversal phenotype in other mutants, including *zar1*, *figla*, *vasa*, and *rbpms2*, that disrupt either early germ cell development or the beginning of female meiotic entry (*Bertho et al., 2021*; *Hartung et al., 2014*; *Kaufman et al., 2018*; *Miao et al., 2017*; *Qin et al., 2018*). Our *foxl2l* mutation results in the disruption of early germ cell development, and *tp53*$^{-/-}$*foxl2l*$^{-/-}$ double mutant fails to rescue the all-male phenotype (*Yang et al., 2025*). In addition to inducing apoptosis, p53 also regulates the differentiation of airway epithelial progenitors (*McConnell et al., 2016*), embryonic stem cells (*Lin et al., 2005*), and brown adipocytes (*Molchadsky et al., 2013*). Thus, p53 may participate in the alternative differentiation of mutant germ cells. We have also identified an uncharacterized gene, *zgc:158463*, which is specifically expressed in *foxl2l* mutant cells in I3 and I6. Its high expression suggests that *zgc:158463* may be involved in the male differentiation of mutant cells. This assumption still needs to be tested by more functional studies.

# Materials and methods
## Animals

WT zebrafish *Danio rerio*, TL and Nadia strains, *piwil1:egfp* transgenic (*Pan et al., 2022*) and *foxl2l*$^{-/-}$ mutant fish were maintained in system water (28.5°C, pH 7.2, conductivity 500 µS/cm) according to the standard protocol (*Westerfield, 2000*). For embryos, fertilized eggs were raised in 100 mm Petri dish with 100 eggs per dish until 4 dpf. Larvae were then transferred to a 1 l beaker with 60 larvae in 600 ml system water and fed with paramecia until 12 dpf. At 8 dpf, larvae density was reduced to a half and fed with brine shrimps twice daily. At 21 dpf, larvae were transferred to the system tank.

For sample collection, the growth of larvae was determined according to both age and body length (**Nüsslein-Volhard and Ralf, 2002**). Zebrafish were handled under the guidelines of the Institutional Animal Care and Utilization Committee of Academia Sinica (IACUC: 10-10-084) and National Laboratory Animal Center of National Applied Research Laboratories (IACUC: NLAC-113-M-004-R1).

## Generation of *foxl2l* mutant

The gRNA sequences targeting the FH domain of *foxl2l* were designed using CHOPCHOP (https://chopchop.cbu.uib.no) and CRISPRscan (https://www.crisprscan.org). Target sites with high efficiency, low off-target effect, and the sequence of NGG at the 3' end were chosen. To get higher knockout efficiency, two gRNAs with 50 pg each and 200 pg of TrueCut Cas9 Protein v2 (Life Technologies, Cat. No. A36498) were co-injected into the embryo at the two-cell stage. F0 fish were crossed with WT fish to generate stable mutant lines. To determine the genotype, the fin of F1 offspring was heated in 50 mM NaOH at 95°C followed by the addition of 1/10 volume of 1 M Tris-HCl pH 8 on ice for 5 min before DNA amplification by PCR (F, 5'-AGTAAACCTGAAGCACACCTGG-3', R, 5'-CATCCCTT TTTGTTCTTCTCGT-3') using Fast-RunTM 2x Taq Master Mix without Dye (Protech Technology Enterprise CO., Cat. No. PT-TMM228-D). The size difference between WT and mutant alleles was determined from capillary electrophoresis, run by QIAxcel DNA Screening Kit (QIAGEN, Cat. No. 929004), detected by High Performance Nucleic Acid Analyzer (eGENE HDA-GT12) and analyzed by QIAxcel BioCalculator software.

## RNA analysis and plasmid

Tissue RNA was extracted by TRIzol Reagent (Invitrogen, Cat. No. 15596026). cDNA was reverse-transcribed by Maxima First Strand cDNA Synthesis kit (Fermentas Int, Cat. No. K1641) from 1 µg of RNA.

For probe synthesis, templates were either constructed into vector (for *foxl2l*, *nanos2*, and *id1*) or amplified by PCR (for *rec8a*, *obf3b*, and *tekt1*). For construction, cDNA form ovary (for *foxl2l* and *nanos2* probe) or 24 hpf larvae (hours post fertilization) (for *id1*) (Addgene) were amplified by PCR (*foxl2l*: F, 5'-CTTTCCACCTGTACCGTGCG-3', R, 5'-CAGTCAGCACCGAGGTTTGC-3'; *nanos2*: F, 5'-GACGGATCCCATGGGCAAAACACACCTAAAACA-3', R, *id1*: F, 5'-TGGTGAACTGTCATCGCACT-3', R, 5'-AGCGTTCACATCATATGGCA-3') using Taq DNA polymerase (Roche, Cat. No. 11146165001). Fragments of *foxl2l* and *id1* were cloned into *pGEMT-easy* plasmid by TA cloning, while the *nanos2* fragment was cloned into pCS2⁺ plasmid by *BamHI* and *KpnI*. Plasmid was cut, and the antisense probe was synthesized by in vitro transcription with Dig RNA Labeling Mix (Roche, Cat. No. 11277073910) or Fluorescein RNA labeling mix (Roche, Cat. No. 11685619910) together with T7 RNA polymerase (Roche, Cat. No. 10881775001), SP6 RNA polymerase (Roche, Cat. No. 10810274001), or T3 RNA polymerase (Roche, Cat. No. 11031163001) (*foxl2l: NcoI*, SP6; *nanos2: BamHI*, T3; *id1: SacII*, SP6). For *rec8a*, *obf3b*, and *tekt1*, cDNA containing T7 promoter was amplified by PCR (*rec8a*: F, GAGTATTT AGGTGACACTATAGACAATTCCCCCTCAGCAACC, R, GAGTAATACGACTCACTATAGGGGATGC ACCGGTGATTTGTGC; *obf3b*: F, 5'-GAGTATTTAGGTGACACTATAGGGGGCAACTGGAATGAAT AA-3', R, 5'-GAGTAATACGACTCACTATAGGGACTACGACCGCTGAAGGAGA-3'; *tekt1*: F, 5'-GAGT ATTTAGGTGACACTATAGGGAGGATCCAGGACATCAAA-3', R, 5'-GAGTAATACGACTCACTATA GGGCCTTCTCGGCTTTGCTAATG-3') using Fast-RunTM 2x Taq Master Mix without Dye (Protech Technology Enterprise CO., Cat. No. PT-TMM228-D). After purification by QIAquick PCR Purification Kit (QIAGEN, Cat. No. 28106), antisense probe was synthesized by in vitro transcription as described above.

For the generation of *foxl2l* mutant lines, the *pT7-gRNA-foxl2l#1* and *pT7-gRNA-foxl2l#2* plasmids containing oligo against two different target sites of *foxl2l* were constructed. After annealing oligo pairs (*foxl2l#1*: F, 5'-TAGGAGCTGGATGAATGAAACG-3'; R, 5'-AAACCGTTTCATTCATCCAGCT-3'; *foxl2l#2*: F, 5'-TAGGAGCCACGTACGAATAAGG-3', R, 5'-AAACCCTTATTCGTACGTGGCT-3') *foxl2l#1* and *foxl2l#2* fragments were cloned into *pT7-gRNA* (Addgene) through one-step digestion and ligation, respectively. Reagents (400 ng of *pT7-gRNA*, 0.25 µM annealed oligo), 1x NEBuffer 3.1 (New England Biolabs, Cat. No. B7203), 1x T4 DNA ligase buffer (Promega Co., Cat. No. C126B), 1.5 U T4 DNA ligase (Promega Co., Cat. No. M180A), 5 U *BsmBI* (New England Biolabs, Cat. No. R0580), 3 U *BglII* (New England Biolabs, Cat. No. R0144S), and 6 U *SalI* (New England Biolabs, Cat. No. R0138S) were incubated with the following condition: 20 min in 37°C and 15 min in 16°C for three cycles,

followed by 10 min in 37°C, 15 min in 55°C, 15 min in 80°C, and cooling in 4°C, for plasmid construction. Then, the plasmids were linearized by *BamHI*, and gRNA was synthesized by MegaShortscript T7 Transcription Kit (Thermo Fisher Scientific, Cat. No. AM1354).

## RNA in situ hybridization

Whole-mount in situ hybridization is performed using a published protocol (*Thisse and Thisse, 2008*). Larvae or tissues were fixed in 4% paraformaldehyde (PFA) overnight at 4°C before the removal of head, tail, and intestine. The dissected samples went through dehydration, rehydration, permeation, hybridization, antibody incubation, and staining. For permeation, larvae were treated with 10 µg/ml Proteinase K (Roche, Cat. No. 03115887001) for 10 min while adult tissues were treated for 40 min. For probe hybridization, samples were incubated with 1 ng/µl digoxigenin (DIG)-labeled probe at 70°C overnight. For color staining in situ hybridization, the antibody incubation was performed with 1:5000 diluted Anti-DIG-AP antibody (Roche, Cat. No. 11093274910) at 4°C overnight. Samples were stained with BM-purple (Roche, Cat. No. 11442074001). FISH was performed as described (*Brend and Holley, 2009*) with 555 Styramide Kit (AAT Bioquest Inc, Cat. No. 45027) and 488 Tyramide Conjugate (Biotium Inc, Cat. No. 92171) followed by DAPI staining with 1:20,000 dilution. For photography, samples were mounted in 85% glycerol.

## Photography

Photography was performed using AxioImager Z1 upright microscope (Carl Zeiss Inc) with AxioCam HRc digital camera. Image was processed by Axiovision 4.7 software (Carl Zeiss Inc). For fluorescent samples, photography was performed by LSM 780 Confocal Microscope (Carl Zeiss Inc).

## Mating test

Adult females with different genotypes were mated with WT males individually from 4 months of age. Ten matings were performed with 7- to 14-day intervals. To avoid individual differences, the first three matings were considered as practice. Only the following seven matings were included in the experimental counting. Fertilization rate was calculated as the number of fertilized eggs/total number of eggs at 24 hpf. Each experimental group contained three to seven mating pairs.

## Histology

Larvae, ovary, and testis were fixed in Bouin's solution at 4°C overnight before six washes in phosphate-buffered saline (PBS) for 30 min each, serial dehydration from 50% to 100% ethanol and incubation with xylene and paraffin twice each. Then, the samples were embedded in paraffin before being sectioned with 6 µm thickness. The sectioned samples were stained with hematoxylin and eosin for histological observation.

## Immunofluorescence

Larvae were fixed in 4% PFA overnight before removal of head, tail, and intestine. The dissected samples went through three washes of PBS with 0.2% Triton X-100 (PBST) for 5 min each, blocking with normal goat serum for 1 hr at room temperature, incubating with primary antibody (Vasa [GeneTEX, Cat. No. GTX128306, RRID:AB_2847856], 1:200 dilution; Sycp3 [Abcam, Cat. No. ab150292], 1:100 dilution) 4°C overnight, three washes in PBST for 30 min each and incubating with 1:200 diluted secondary antibody (Alexa Fluor 546 Donkey anti-Rabbit IgG [Invitrogen, Cat. No. A-10040, RRID:AB_2534016]; Alexa Fluor 488 Goat anti-Rabbit IgG [Invitrogen, Cat. No. A-11034, RRID:AB_2576217]) together with 1:20,000 diluted DAPI. Finally, the samples were mounted in 85% glycerol for photography.

## Transcriptome analysis of scRNAseq

Germ cells from 26 dpf WT or *foxl2l*[10i/10i] zebrafish in *piwi1:EGFP* transgene background were collected for scRNAseq. At this juvenile age, fish do not have apparent sexual characteristics. To isolate EGFP+ germ cells, the trunk of 20 fish was collected and dissociated in 0.4% collagenase (Worthington Biochemical, Cat. No. LS004188) and 0.5% trypsin (Life Technologies, Cat. No. 15400-054) in 1 ml PBS at 28°C for 60 min. After dissociation, cells were supplemented with fetal bovine serum (FBS) (Life Technologies, Cat. No. 10437028), centrifuged at 600×*g*, 4 min, and the pellet was suspended in FACS Pre-Sort Buffer (BD Biosciences, Cat. No. 563503) before being filtered through

cell strainers. 1 mg/1 μl of propidium iodide (Sigma-Aldrich, Cat. No. P4170) was added to the eluate to stain dead cells, and around a 75% survival rate was obtained in both genotypes. EGFP⁺ cells were collected into 60% Leibovitz's L-15 medium (Thermo Fisher Scientific, Cat. No. 11415064) by FACSAria III Cell Sorter (BD Biosciences) and supplemented with 10% FBS. Total 4781 cells for *foxl2l*⁺/⁺ and 8673 cells for *foxl2l*¹⁰ⁱ/¹⁰ⁱ samples were obtained upon counting by a cell sorter. After centrifugation at 300×*g* for 5 min and resuspension in PBS with 0.04% bovine serum albumin (Sigma-Aldrich), the recovered cell number was 3231 in *foxl2l*⁺/⁺ and 4019 in *foxl2l*¹⁰ⁱ/¹⁰ⁱ. Finally, the single-cell cDNA libraries were constructed by Chromium Next GEM Single Cell 3' Library Construction Kit v3.1 kit (10x Genomics) following the manufacturer's protocols with 12 cycles for cDNA amplification and 14 cycles for library amplification. Sequencing was performed by NextSeq 500 (Illumina) with 150 cycles of 28×91 paired-end reads. The raw base call (BCL) files were transferred to FASTQ files by the cellranger mkfastq pipeline from Cell Ranger (10x Genomics). The reads were mapped to the zebrafish genome GRCz11, and gene expression was obtained by cellranger count from Cell Ranger (10x Genomics).

## Statistical analysis

All the quantitative data is presented as mean ± SD. The data was analyzed by Student's t-test using GraphPad Prism 8.

## Single-cell transcriptome analysis

### Count matrix generation

Raw count matrices were generated by Cell Ranger v3.1.0 (*Zheng et al., 2017*) using the zebrafish reference genome GRCz11 and reference transcriptome ensemble release v101. The count matrices were then stored as Seurat objects v3.2.3 (*Stuart et al., 2019*).

### Quality control

A two-step quality control approach was conducted individually for each sample. First, we conducted quality control at the cell level. Cells with UMI counts <10,000 or mitochondrial gene expression percentages ≥10% were marked as low-quality cells. Multiplet detection was performed on the remaining cells using the R package *scds* (*Bais and Kostka, 2020*). Cells with a hybrid score ≥ the outer fence were marked as multiplets. Second, we performed quality control at the cluster level to screen out the potential low-quality cells that passed the hard thresholds at the cell level. A low-resolution clustering was performed using Seurat v3.2.3. Clusters containing more than 80% low-quality cells or multiplets were marked as low-quality clusters. Together, cells that were marked as low quality, multiplets, and cells in low-quality clusters were excluded from the subsequent analysis.

### Data preprocessing and cell-cycle scoring

The count matrices of the remaining 769 cells from the WT and 2399 cells from the mutant were merged and normalized using the R package *scran* (*Lun et al., 2016*). After normalization, gene expression data of cells from WT and mutant samples were organized into individual Seurat objects. The function *CellCycleScoring* in Seurat v3.2.3 was used for cell-cycle scoring (*Figure 3—figure supplement 3*). Briefly, all genes were first sorted by their mean expression and split into bins. For each S- and G2M-specific gene, relative expression was calculated by comparison with other genes in the same bin. The S-score and G2/M-score were calculated as the average of the relative expression of all genes in the lists of S- and G2M-specific genes.

### Cell clustering analysis of WT cells

We used the *FindVariableFeatures* in Seurat v3.2.3 to identify the top 2500 highly variable genes in each sample. The principal components (PCs) of these highly variable genes on scaled expression matrices were derived after their cell-cycle scores were regressed out (*Figure 3—figure supplement 4*). We used the leading 30 PCs for UMAP embedding. The Leiden algorithm was applied to identify cell clusters. The optimal resolution, determined by the highest average silhouette score, was chosen from a range of feasible values.

## Trajectory and pseudo-time analysis of WT cells

The data matrix, along with the cell clusters and UMAP embedding, was converted to the CDS format and subsequently analyzed using Monocle 3 (*Cao et al., 2019b*) to obtain the trajectory that encompasses all clusters. We then used the GSC marker *nanos2* to identify the root and the order of the trajectory and to estimate the pseudo-time for each WT cell.

## Cluster marker identification

We used the *top_markers* in Monocle 3 (*Cao et al., 2019b*) to identify the cluster markers whose high expression discriminates cells in the target cluster from the other cells with a high degree of specificity. For each cluster, genes that satisfy the following criteria: (i) specificity ≥0.2, (ii) q-value <0.01, and (iii) having nonzero counts in ≥80% cells in the cluster were selected as markers. Marker genes were ranked based on the specificity within WT clusters, and the top 2–4 markers in each WT cluster were selected as top markers. Total 37 top markers were shown in *Figure 1D*.

## Inferring the developmental stages to mutant cells

The function *TransferData* in Seurat v3.2.3 was utilized to infer the developmental stage from the WT cells to the mutant cells. Initially, PCs were derived from the WT cells based on the expression of their cluster markers. Mutant cells were then projected to the same PCs. The PCs and the projected PCs were used to identify the mutual nearest neighbors across mutant cells and WT cells. These mutual nearest neighbors were considered anchors and scored based on the consistency across the neighborhood structure of each dataset. Lastly, the developmental stage of each mutant cell was determined using a weighted voting mechanism. This process considers the stages of WT cells in every anchor, with votes weighted according to the anchor score and the proximity between the mutant cell and its corresponding anchor mutant cell.

## Co-clustering analysis of mutant cells and WT cells from GSC to Prog-C

To integrate mutant cells and WT cells from GSC to Prog-C, the function *DataIntegration* in Seurat v3.2.3 was applied. Briefly, both datasets were projected to a shared space by performing canonical correlation analysis. Mutual nearest neighbors were then identified as anchors, representing pairs of mutant and WT cells sharing a similar expression pattern. Anchors were scored based on the consistency across the neighborhood structure of each dataset. For each WT cell, a correction vector was created based on the proximity to every anchor and the anchor score. The WT expression matrix was corrected accordingly and integrated with that of mutant cells for the same clustering procedure as previously described. Mutant-specific marker genes were identified using the top_markers function with the following criteria: (i) specificity ≥0.25, (ii) Q-value <0.01, (iii) presence in ≥80% mutant cells for each integrated group, and (iv) not included in the WT cluster marker gene list.

## Composition analysis

Because each stage has its own overall rate of prevalence in the entire cell population, we adjusted for the unevenness of stage prevalence in determining how prominent a stage is in an integrative group. Fisher's exact test was used to find out which integrative groups were enriched with what stages of cells. Prominent stages for an integrative group are defined as those with significantly higher proportions of cells than their prevalence rates. Specifically, we calculated the proportions of developmental stages for both WT and mutant cells in each group and then conducted Fisher's exact tests for each stage within each group. The resulting p-values were adjusted by the Benjamini-Hochberg method for controlling for false discovery rates, which were set at 0.05. The stages with significant enrichment are referred to as the prominent stages after prevalence rate calibration, or prominent stages for short.

## Differential expression analysis

To identify DEGs during development, we compared gene expression levels between mutant and WT cells at the same prominent stage within each integrated group. Other cells not included in the comparison are shown in gray on UMAP (*Figure 4A*). The Wilcoxon rank sum test was performed using the function *FindMarkers* in Seurat v3.2.3. Genes that satisfy the following criteria: (i)

|log2(fold-change)|≥log2(2.5) and (ii) q-value<0.05 were selected as DEGs. The DEGs in each stage were ordered by the fold change.

## Reprocessing of the public scRNAseq dataset from Liu et al

The processed data were retrieved from the designated repository (GEO accession number: GSE191137). To visualize the expression of GSCs and progenitors of WT cells in 40 dpf, we selected a subset of cells with sufficient UMI counts (≥7000) from clusters 0 and 2 and lacked *zp3.2* gene expression. After regressing out the cell-cycle scores, we used the leading 30 PCs for UMAP embedding.

## Acknowledgements

We would like to thank the help of Flow Cytometry Core (Institute of Biomedical Sciences), Genomics Core and Bioinformatics Core (Institute of Molecular Biology Academia Sinica) in scRNAseq study, and Taiwan Zebrafish Core Facility (NSTC 112-2740-B-400-001), Imaging Core (Institute of Molecular Biology Academia Sinica) for confocal imaging. This work was funded by grants from National Health Research Institutes (NHRI-EX107-10506SI) and National Science and Technology Council (NSTC 113-2313-B-492-001-, MOST 108-2311-B-001-038-MY3) to BcC.

## Additional information

### Funding

| Funder | Grant reference number | Author |
|---|---|---|
| National Health Research Institutes | NHRI- EX107-10506SI | Bon-chu Chung |
| National Science and Technology Council | MOST 108-2311-B-001-038-MY3 | Bon-chu Chung |
| National Science and Technology Council | NSTC 113-2313-B-492-001- | Bon-chu Chung |

The funders had no role in study design, data collection and interpretation, or the decision to submit the work for publication.

### Author contributions

Chen-wei Hsu, Conceptualization, Data curation, Formal analysis, Validation, Investigation, Visualization, Methodology, Writing – original draft, Writing – review and editing; Hao Ho, Resources, Data curation, Software, Formal analysis, Validation, Investigation, Visualization, Methodology; Ching-Hsin Yang, Data curation, Validation, Investigation, Methodology, Writing – review and editing; Yan-wei Wang, Data curation, Formal analysis, Investigation, Methodology, Writing – review and editing; Ker-Chau Li, Resources, Software, Formal analysis, Supervision, Investigation, Visualization, Methodology, Project administration, Writing – review and editing; Bon-chu Chung, Conceptualization, Formal analysis, Supervision, Funding acquisition, Investigation, Writing – original draft, Writing – review and editing

### Author ORCIDs

Chen-wei Hsu  https://orcid.org/0009-0002-8845-7920
Hao Ho  https://orcid.org/0000-0001-8911-7907
Bon-chu Chung  https://orcid.org/0000-0002-8612-0219

### Ethics

Zebrafish were handled under the guidelines of Institutional Animal Care and Utilization Committee of Academia Sinica and National Center for Biomodels, National Applied Research Laboratories (IACUC: 10-10-084, and NLAC-113-M-004-R1).

### Decision letter and Author response

Decision letter https://doi.org/10.7554/eLife.100204.sa1
Author response https://doi.org/10.7554/eLife.100204.sa2

## Additional files

### Supplementary files

MDAR checklist

Supplementary file 1. The list and the expression of marker genes in each cluster.

### Data availability

Data for single-cell transcriptomic data analysis are available with NCBI GEO (https://www.ncbi.nlm.nih.gov/geo/query/acc.cgi?acc=GSE173718). Analysis code is available at Figshare (https://doi.org/10.6084/m9.figshare.26314126.v1).

The following datasets were generated:

| Author(s) | Year | Dataset title | Dataset URL | Database and Identifier |
|---|---|---|---|---|
| Chung BC | 2025 | Single cell transcriptomes of zebrafish germline reveal progenitor types and feminization by Foxl2l | https://www.ncbi.nlm.nih.gov/geo/query/acc.cgi?acc=GSE173718 | NCBI Gene Expression Omnibus, GSE173718 |
| Ho H, Hsu C, Yang CH, Wang Y, Li KC, Chung B | 2025 | Codes of "Single-cell transcriptomes of zebrafish germline reveal stemness suppression and progenitor feminization by Foxl2l" | https://doi.org/10.6084/m9.figshare.26314126.v1 | figshare, 10.6084/m9.figshare.26314126.v1 |

The following previously published dataset was used:

| Author(s) | Year | Dataset title | Dataset URL | Database and Identifier |
|---|---|---|---|---|
| Liu Y | 2021 | Single-cell transcriptome reveals insights into the development and function of the zebrafish ovary | https://www.ncbi.nlm.nih.gov/geo/query/acc.cgi?acc=GSE191137 | NCBI Gene Expression Omnibus, GSE191137 |

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
