## [Editor Report]

Findings have important significance for understanding the role of the *Foxl2*-family of proteins in the early stages of germ cell maturation. Evidence to support the conclusions are solid and use state-of-the-field single cell transcriptomics. Results will interest biologists investigating mechanisms of reproduction, causes of infertility, and the mechanisms by which gonads embark on becoming a testis or an ovary.

---

## [Decision Letter]

**Decision letter after peer review:**

Thank you for submitting your article "Single cell transcriptomes of zebrafish germline reveal stemness suppression and progenitor feminization by Foxl2l" for consideration by *eLife*. Your article has been reviewed by 3 peer reviewers, and the evaluation has been overseen by a Reviewing Editor and Didier Stainier as the Senior Editor.

Essential Revisions:

1) Evidence that committed progenitors revert to germline stem cells (GSCs) is weak. Functional data are required to draw this conclusion. The phrase 'reveal stemness suppression' should be removed from the title and alternative explanations should appear in the text. The claim that suppressing stemness is necessary for female differentiation is not well supported, for example, cells might initiate the committed progenitor program but fail to shut off and continue to express part of the GSC program.

2) Essential literature that places the work in context must be discussed, with several cases noted in each specific review.

3) Alternative explanations to the statement that committed progenitor stage is "the gate toward female determination" must be discussed, for example, that the late progenitor stage is the key stage for female determination.

4) Consider alternatives to the conclusion that the progenitors are differentiated because it is based solely on the expression of foxl2l, which is initially expressed in the juvenile ovary state that lab strains develop through.

5). Consider alternative explanations for the differences in the action of *Foxl2*(Fox3) in zebrafish and medaka.

6) Methods require more details on how fish were selected for scRNA-seq.

*Reviewer #1 (Recommendations for the authors):*

Specific points.

1) Page 7 lines 258-261. The authors conclude that the committed progenitors revert to GSCs based on the coexpression of nanos2 and foxl2l nanos2 and later on the same page line 262, based on expression of id1 in mutants but not WT. Without functional data demonstrating that the progenitors revert to an early state, alternative interpretations should be considered. For example, can the authors exclude the alternative possibility that the cells initiate the committed progenitor program but do not shut off and continue to express the GSC program such that the coexpression of both programs blocks differentiation? It would help to discuss these findings in the context of Fox family functions. Some Fox family members, *Foxl2* and FoxPs for example, are known to be both activators and repressors of transcription or act primarily as repressors. Relevant to this work, repressive activity of *Foxl2* has previously been reported in the mammalian ovary (Pisarska et al. Endocrinology 2004, Pisarska Am J. Phys Endo. Metabolism 2010, Kuo Reproduction 2012, Kuo Endocrinology 2011, as well as several more recent publications). In this context interfering with *Foxl2* repression was proposed to accelerate follicle recruitment and cause premature ovarian failure.

2) Discussion Page 9 lines 289-293. This section is not entirely clear. First GSC-I and GSC-II subtypes are introduced, then it is stated that "all GSCs self -renew" and a Saito 2007 paper is cited. This is followed by a description of "two types of daughter cells derived from GSC self-renewal". Does the conclusion that "all GSCs renew" still hold in light of the two GSC populations? Would a division that produces a daughter that cannot renew be considered "self-renewal" or instead a non-renewing division?

3) Discussion Page 9 Line 295-7. The authors conclude that the committed progenitor stage is "the gate toward female determination" and that they "stay at S-Phase temporarily before differentiation". Is this conclusion based solely on RNA expression? Also, in several species, including zebrafish, meiotic entry has been correlated with ovary development. Can the possibility that the late progenitor stage, the stage when meiotic genes are detected and a stage missing in foxl2l mutants, is the key stage for female determination be excluded?

4) The same comment as above pertains to Page 9 lines 281-284 of the discussion.

5) Page 3 Line 64 states that "the developmental stages of germ cells in zebrafish remain unclear". This is somewhat misleading as recently published single cell datasets have analyzed and defined the stages of germ cell development, including during and after sexual differentiation. This should be clarified, and the relevant published work should be cited. (Liu et al. *eLife* 2022, Wilson et al. Front Cell Dev Bio 2024.)

6) Page 3 Line 70 might be confusing because it states that "XX medaka fish with foxl3 mutation become hermaphroditic processing functional sperms". Please clarify whether they also produce oocytes or are genetically female "XX" but make sperm.

7) On Page 9 lines 305-307, the authors discuss prior working showing that loss of germ cells leads to male development and that germ cells are required for female development. Then lines 307-309 state that this work shows some progenitors are already sexually differentiated. First, the stages compared are completely different. The early work looks at the primordial germ cells and their loss in the first few days of development before a gonad forms and this work looks after a gonad has formed and when sex is being determined. The second concern is that the conclusion that the cells are differentiated is based on the expression of foxl2l, which is expressed in the juvenile ovary state that lab strains have been shown to develop through (Wilson et al. Front Cell Dev Bio 2024). While it is fair to state that some cells express ovary markers, are they really differentiated? For example, in the context of the foxl2l mutant, GSCs and early progenitors inappropriately express foxl2l, but the mutants develop as males. Please clarify.

8) Page 9 lines 315-318 compares medaka and zebrafish foxl2l mutants and seems to suggest that Foxl2l is required for meiosis in medaka but has a different role in zebrafish. Is it possible that foxl2l plays a similar role in repressing the earlier developmental programs of GSCs and early progenitors in both cases? Is it possible that continued expression of these early programs interferes with activation of meiotic genes? Might this account for the absence of the late progenitor stage in foxl2l mutants since it is defined by expression of foxl2l and meiotic genes?

9) Page 10 lines 328-330. Direct differentiation of testis without going through a female phase is discussed. Recent work showing direct differentiation in wild strains should also be discussed and cited (Wilson et al. Front Cell Dev Bio 2024).

10) Page 10 lines 359-360. The authors state "Foxl2l may ensure female differentiation by preventing stemness and antagonizing male development." It is unclear why suppressing stemness would be necessary for female differentiation since female zebrafish have stem cells as do male zebrafish. It seems likely that turning off the GSC and early differentiation programs is important for allowing expression of meiosis and oocyte differentiation genes.

11) Page 11 370-374. "Thus, p53 may assist alternative differentiation of mutant germ cells". What is the evidence that p53 is involved in differentiation of germ cells? Sex bias has not been associated with the published p53 mutants. Is it possible that p53 eliminates germ cells that are simultaneously expressing programs of multiple stages? While p53 has been shown to be important for ovary to testis transformation in mutant contexts in adults, it is dispensable in mutants that disrupt ovary development at earlier stages. Please discuss in the context of, and cite, the relevant ovary literature Rodriguez-Mari et al. PLoS Gen 2010, Shive PNAS 2010, Hartung et al. Mol. Reprod. Dev 2014, Miao Development 2017, Kaufman et al. PLoSGen 2018, Bertho et al. Development 2021.

12) Figure 2D. It would be helpful to show the individual fish points or to present the data as a violin plot.

13) Figure 2F. The numbers are low here only 3-4 gonads are examined for each genotype and marker and in the mutants only 1/4 expresses the "testis" markers.

14) Figure 6 model. What is the evidence that prog C becomes GSCs again/revert versus continuing to express earlier programs?

*Reviewer #2 (Recommendations for the authors):*

The evidence that foxl2l cells are arrested at the committed progenitor stage (a possible novel phenotype) is based on the absence of expression of rec8a and Sycp3. Another explanation is that these gonads have already differentiated as testes. Is rec8a also expressed in male germ cells or is it female germ cell specific? Germ cells in testes are usually delayed in entering meiosis (and thus expressing Sycp3) relative to females so it is not clear if this represents the expected trajectory of testis vs. ovary or a novel developmental state caused by absence of foxl2l. Was expression of any male-specific genes identified in the mutant data set (e.g. tekt1 or odf3b) which could indicate these germ cells were already committed to spermatogenesis? Also, by "arrested" do the authors mean that they are in a cell cycle arrest? If so, this can be assayed with BrdU or EdU labeling.

L99: incomplete sentence.

L128: The publication associated with this dataset should be referenced (it's referenced in the figure legend but should also be referenced in main text).

L138: Using a negative result (absence of staining) to argue for a particular cell stage (prog-E) is problematic. To bolster this argument the authors should consider pointing out that this result is consistent with the scRNA-seq data that predicts little to no overlap between nanos2 and foxl2l expressing cells. Perhaps adding the nanos2 and foxl2l and rec8a UMAPs to Figure 1 would help.

L148 and Figure 2A. Wilson et al. (2024) showed that the gonads of Nadia ZZ fish appear to directly develop into testis without producing oocytes during the bipotential stage. The data in Figure 2A showing foxl2l expression in the ZZ gonads is therefore unexpected. Can the authors comment on this?

L154: "…barley observed in testis." Should this be "…not observed…", or is there low but detectable expression of foxl2l in the testis (not clear from Figure 1D)?

L170: It is not clear from the Methods how the fish were selected for scRNA-seq. With the piwil1:egfp line it is possible to distinguish wild-type males from females at the 27 dpf timepoint used in this analysis based on the size of the gonad (large=female, small=male). Were only fish with large gonads used for WT and foxl2l mutant scRNA-seq? If fish were chosen randomly, it would be expected that germ cells from testes would also be included. When do foxl2l mutant gonad first start looking like testes? Is there a delay in testis development relative to wild-type?

Figure 2B: Including age-matched wild-type testes in Figure 2B may help clarify if the foxl2l gonads are different or similar to a normal male gonad at this stage of development.

Figure 2D: I would suggest replacing si:ch211-191j22.7 with meiosin, since this is the clear ortholog.

Figure 3B: It is hard to see the morphology of the nuclei in these images. Grey scale is perfect for the DAPI channel, but it would be helpful to increase the apparent signal intensity using an image processing program like Photoshop so that it is possible to better assess the morphology of nuclei.

Figure 3C: I am having a hard time understanding the meaning of Figure 3C. For the WT cells, the nanos2+ cells are restricted to the W1 subcluster (Figure 1D), indicating that the GSCs are homogeneous with respect to gene expression. However, when WT cells are clustered with mutant cells the WT GSCs now partition to two different subclusters (I1 and I2). When the "control" cells are not clustering tightly, it calls into question how much this clustering represents reality. How do the authors rationalize this?

L194: Figure S7B is unfortunately not helping me understand how the data was processed. What is meant by "transfer developmental stage?" Does this require/assume that the mutant cells be a specific stage that can be mapped directly onto wildtype stages? I wonder if forcing the mutant cells to cluster with wildtype cells is somehow skewing the results. How different is the clustering if only mutant cells are used?

Figure 3E: nanos2 and cbx7a are listed twice. It is likely they should be psmb13a and dnmt3bb.1, respectively. Was expression of any male-specific gene identified in the mutant data set (e.g. tekt1 or odf3b) which could indicate these germ cells were already committed to spermatogenesis?

Sup Figure S8A: It would be helpful if this figure also included the UMAP shown in Figure 3C for reference (with I1-I7 labels). S8A is hard to interpret given the altered gene expression pattern of mutant cells. Some genes fit the proposed cell stages (e.g. dnmt3bb.1), but others are not as clear (e.g. nanos2, dhrs1).

L210: "This suggests that GSCs can be further divided into two finer stages, GSCI and GSCII." If you only see this subdivision in mutants, then it is hard to argue that these are representative stages present in WT.

Line 307: It is not clear what this statement means. It appears to imply that foxl2l-expressing cells are sexually differentiated. However, in Figure 2A it is shown that foxl2l is expressed in germ cells of ZZ fish during the bipotential stage, yet these germ cell presumable would have differentiated as sperm, not oocytes.

L302-322: The authors need to spell out the evidence that supports these statements. It could be argued that the difference is not in the role of foxl2l between Medaka and zebrafish, but the fact that medaka has a strong genetic sex determination mechanism while zebrafish does not. Thus, the apparent difference in phenotype is because medaka cannot sex reverse as easily as zebrafish.

L331: For completeness, please add a reference to Wilson et al., (2024; PMID: 38529407)

L356: The authors argue that one role of Foxl2l is to suppress *Dmrt1* expression in progenitor cells, yet in the foxl2l mutant germ cells, *Dmrt1* is upregulated in all mutant cell stages, from GSC to prog-C cells (i.e. even in cells that do not normally express foxl2l expression). It is possible that the upregulation in all cells is because the gonads from which these cells were isolated have already begun to differentiate as a testis and the germ cells are now more male-like than female-like. This would also suggest the possibility that male and female GSC's do not express equivalent amounts of *Dmrt1*.

*Reviewer #3 (Recommendations for the authors):*

The major conclusion, i.e., title and short title may be misleading and may cause confusion. Stemness is the ability of a cell to perpetuate/continue/preserve its lineage, to give rise to differentiated cells. With foxl2l, progenitor cells committed cell (Prog-C) will continue to develop into late progenitor (Prog-L). In addition, there is no strong evidence that Prog-C was able to dedifferentiate back into GSC in foxl2l mutants. Prog-C could be simply lost in foxl2l mutant, and increased nanos2 cells could be due to increased proliferation of GSC. Therefore, "Foxl2l suppresses stemness" could be misleading. Is there size and morphological difference between prog-C and prog-L?

Lines 23-24, 64-65 and in other parts of the text "germ cell types … remain elusive" "developmental stages of germ cells in zebrafish remain unclear." Probably should be specified as early germ cell types. Five developmental stages of germ cells (oocytes) in zebrafish are well defined by Kelly Selman since 1993. In fact, this manuscript was focusing on fine classification and separation of different germ cell types prior to stage I, which should be highlighted or emphasized, because this is not possible until scRNAseq became available.

The label D for figure 1D is too far, should be close to the sub-panel.

It is hard to view two different symbols illustrated in Figures 3C and 4A, suggest to separate symbols into side-by-side figures.

The cell numbers for mutant in figure 3D are much higher than those in WT, why?

Why are there two nanos2 lines (lines 1 & 3) in the same GSC group with different expression profiles in Figure 3E? nanos2 expression was not listed for the mutants in figure 3E. Y-axis had label for mutant, but does not have the label for WT, not was explained in the figure legend.

Lines 171-173, without foxl2l, which is a transcriptional factor, the gene expression would change. So, is it still validate using wildtype marker genes to define developmental stages found in foxl2l. Can different types of early germ cells be separated using scRNAseq data from the foxl2l mutant? Can pseudotime plots be used for the foxl2l mutant? If so, what looks like?

Abnormal increased expression of nonfunctional foxl2l (figure 3F) in foxl2l mutant suggests a feedback mechanism.

Co-expression and increased expression of nanos2/id1 with foxl2l in GSC in foxl2l mutant is interest, as these two genes are expressed separately in WT. Can overexpress of nanos2 and id1 lead to loss of oocytes, or dedifferentiation of oocytes (Prog-C) back to PGCs, and female to male sex reversal?

*Guest Editor (suggestions for the authors)*

The Hsu et al. manuscript presents valuable data that extend earlier single cell transcriptomic data on zebrafish gonads to help us better understand various initial stages of germ cell development in zebrafish and the role of Foxl2l in oogenesis. Data are well presented. Some of the conclusions do not seem to be well supported by the data, or alternatives are sometimes not considered.

The authors investigated development of zebrafish germ cells by isolating Piwil1-expressing cells and performing scRNA-seq in wild types at 26 days post fertilization, which augments published single cell transcriptomic studies of whole zebrafish gonads at 40 dpf, and they performed scRNA-seq experiments on mutants lacking foxl2l activity. These data, validated by in situ hybridization studies, allowed the identification of new stages of germ cell development. Studies showed that foxl2l mutants failed to develop meiotic oocytes, and that mutant germ cells reached a modified committed progenitor stage (Prog-C) but did not become late progenitors (Prog-L). The cells the authors called Prog-C cells in mutants, unlike wild types, expressed some GSC markers, suggesting that they maybe shouldn't be called Prog-C cells in mutants. From this result, the text concluded that committed progenitors then reverted to germ cell stem cells (GSC), which then directly developed into spermatogenic cells, and hence the animals became males, concluding that this was a direct development into males.

Alternative hypotheses exist for these data. The expression of genes normally expressed by GSCs (e.g., nanos2) in what otherwise seem to be committed progenitors was interpreted as a reversion of Prog-Cs to GSCs. An equally possible hypothesis is that the normally GSC-expressed genes never turned off and that the mutant 'Prog-C' never fully developed. Both explanations need to be presented and arguments assessed for and against each of these explanations.

Additionally, the conclusion that there was direct male differentiation in foxl2l mutants without going through the female phase has alternative explanations consistent with the data. This direct male development argument assumes that the Prog-C stage has not already embarked on a female pathway. The text did not demonstrate that Prog-C, or even the earlier stages like Prog-E, had not already begun to be female. The very name 'committed progenitor cells' suggests that these cells are already committed to the female pathway in wild types.

These authors and others have shown that one of the earliest steps in female gonad development in zebrafish is an increase in the number of primordial germ cells compared to zebrafish that will become males, by 14 days post fertilization. The cell division marker analysis reported here suggests that those PGC proliferation steps are already occurring in the cells studied here. So do the 'Proliferation' arrows shown in Figure 6 for both the Prog-E and Prog-C stages. Thus, these stages would already be female-oriented germ cells. And thus, when the reversion of Prog-C towards Prog-C-to-GSC happens, what that really represents is a reversion of female pathway cells towards uncommitted GSCs that then embark on a male pathway, as happens in many zebrafish individuals due to stress or temperature or other features. Thus, the data presented here shows that the male pathway is taken after germ cells have embarked on the female pathway and turned on foxl2l, but subsequently thwarted from becoming a meiotic oocyte due to the lack of Foxl2l in the mutants.

The title (Single cell transcriptomes of zebrafish germline reveal stemness suppression and progenitor feminization by Foxl2l) should be revised because the conclusion that Foxl2l suppresses 'stemness' is not well supported. The data do show that foxl2l mutants have cell types that co-express GSC markers like nanos2 with genes usually expressed later in female germ cell development, like sycp1 and sycp3 (Figure S6), but that does not clearly mean that these mutant cells are in fact stem cells.

The 'allmarkers' files are not included in the supplementary data, making it impossible to see how genes were selected for display. These files must be added to the supplementary data.

Some suggestions for improving clarity in the text and figures.

P2 'factor, Foxl2l, is expressed in the progenitors committed to the ovary fate.'. Consider saying '…in the germ cell progenitors committed to the oocyte fate..'

P2 'Another single-cell profiling of foxl2l-/- germ cells reveals the arrest of early progenitors.'

Text here uses both present and past tenses. For describing the results of experiments done in the past, past tense might be better, as the text often uses in 'mediated mutation of foxl2l produced 100% male fish' and 'in stem cells) was elevated'. So maybe use 'revealed' instead of 'reveals'. And again could say '…early germ cell progenitors.' Just to be precise. Likewise, '…of nanos2 (germ cell stem cell)…'.

In Key Words, consider changing 'stem cell' to 'germ cell stem cell', and 'germ cell progenitor', which might better direct interested folks to this paper. Consider changing 'female' as a keyword, maybe it's too generic, to 'female-to-male sex change', which might get more focused hits.

P2 'increase of nanos2+ germ cell in foxl2l'. Use here plural, 'germ cells'…

P3 'contrast to higher vertebrates,'. Consider saying what specific taxon is meant here rather than 'higher'. Talking about higher and lower animals comes from a pre-scientific view of the scale of life.

P3 'system, but domesticated laboratory strains have no sex chromosome'. Use plural, 'chromosomes'.

P3 'the differentiation of zebrafish gonad. Zebrafish'. Plural, 'gonads'

P3 'of germ cells (nr0b1 mutants) or no germ cell (dnd morphans) develop'. Plural, and spelling, 'or no germ cells dnd morphants

P3 'However, the developmental stages of germ cells in zebrafish remain unclear.'

I think the developmental stages have been clear, but the molecular genetics that moves cells through these stages is unclear, and the authors' work subdivides certain stages.

P3 'and initiates oogenesis in XX gonad (Kikuchi'. Plural, 'XX gonads'

P3 'with foxl3 mutation become hermaphroditic possessing'. Several changes: 'with a foxl3 mutation become hermaphrodites possessing functional sperm'

P3 'and proved that Foxl2l controls the development of committed progenitor,'. Plural, 'progenitors'.

P4 'Germ cells play important roles in zebrafish sex differentiation, but their development is poorly studied.'

Maybe more accurate to say: 'Germ cells play important roles in zebrafish sex differentiation, but we still have much to learn about their development.'

P4 '2002. Fluorescent germ cells were isolated from piwil1:EGFP transgenic fish by cell sorting'

The reader needs to know when piwil1 turns on in germ cell development with respect to the timing of oogonia vs. spermatogonia commitment. We need to know if by using the transgene we miss early but important stages in germ cell development.

Supplementary Figure S2 must have a mechanism for the reader to identify the genes in the rows. They would be too small to read if written on the figure so it should be in an Excel file in the supplement.

P4. 'expression profiles of the top 37 out of a total of 1491 marker genes'

Top based on what criterion? P-adj? P? Log2 fold change? Most specific 2 or 3 per cluster? Or something else? And what defines a 'marker gene'?

P4 'markers (sycp1, sycp3, dmc1, buc and zp3e). Are buc, which helps assemble germplasm, and zp3e, an eggshell component, \involved in meiosis? Consider rephrasing.

Suppl Figure 3 and 4. Although most of the markers show very similar patterns, thus cross-validating both datasets, it looks like for the markers CU467646.3, si:zfos^-1^505d6.3, and zp3e(zgc:171779), that the developmental pathway extended much further in the Liu data than the Hsu data. This isn't a huge concern, but it should be mentioned and discussed if there is a methodological difference that could explain this difference. Maybe it's just the age difference?

P5 'Zebrafish progenitors are round in shape'. Progenitors of zebrafish oocytes…

P5 'et al., 2010). To characterize these three progenitor subpopulations,'

I'm not sure which three are referred to here. When adding in GSC, we get four. Please clarify.

. 'et al., 2010). To characterize the GSC, Prog-E, Prog-C, and Prog-C(S) progenitor subpopulations,'

P5 'hybridization (FISH) and detected GSC as positive'. Make plural GSCs.

P5 Characterization of the three stages of progenitor subtypes using nanos2, foxl2l, and rec8 was useful.

P5 'premature stop codon disrupting the FH domain of Foxl2l'. Spell out 'forkhead'.

Figures 2C and 2D. These would be more impactful if error bars were displayed on the graphs and numbers of trials or numbers of individuals were shown.

P5 'non-meiotic cystic germ cells were found in foxl2l mutant (Figure 2E).'

Plural: mutants

P5 'foxl2l mutant germ cells are arrested at committed progenitor.'. 'foxl2l mutant germ cells are arrested at the committed progenitor stage.'

P5 '(Supplementary Figure S6A). The expression of the 37 top WT marker genes'. Again in a couple of words, explain the definition of 'top' genes more precisely.

P5 'found that these marker genes were expressed at similar stages as they represented for WT cells.'. The text here needs to say that WT and mutant cells were co-clustered. Also, it would be interesting to see how the pseudotime navigates its way through this blob of cells.

Figure 3D. Label the horizontal axis as cluster number. Also, consider changing 'dominant' in the legend to 'predominant'.

P6 'integrated groups (Figure 3D). Dominant stages were obtained'. Define' Dominant stages'.

P6 'calibration against their prevalence rates in the entire cell population.'

Does 'entire cell population' mean all of the mutant cells? or all of the mutant plus WT cells?

P6 'For mutant cells, while groups I5-I8 had two dominant stages,'

If dominant means the one that's present most frequently, then more than one shouldn't be dominant unless there's a tie. So 'dominant' needs a clear definition.

P6 'that WT and mutant cells with high cell-to-cell similarity belong to similar

developmental stage.'

'that WT and mutant cells that have high transcriptome similarity likely belong to the same developmental stage.'

P6 'Groups I3 and I6 are particularly unique.'

'Groups I3 and I6 are unique.' Unique means the only one of its kind. So there can't be gradations of being unique.

Figure 3E. Should the group at the bottom labeled 'Mutant', be called something like 'Mutant-specific cell types'? Because all of the data in the table refer to mutant cells, right?

P6 'mutant cells additionally expressed zgc:158463'

In addition to what? So mutant cells expressed all the genes of WTs in I3 but in addition zgc:158463 which wasn't expressed in the WT cells in I3? The same question for I6 and tp53. Please clarify.

P6 'Furthermore, the Prog-C marker foxl2l was expressed in mutant cells of I1,'

So that means that the mutant mRNA had normal stability? Wasn't degraded by nonsense mediated decay?

P6 'The aberrant foxl2l expression in some earlier GSC and Prog-E plus the accumulation of foxl2l expressing cells in the mutant indicate that progenitor development is impaired in foxl2l mutants.'

'in the mutants and the failure to find clusters representing more mature stages indicate that progenitor development is impaired in foxl2l mutants' [could that be added or not?]

P7 'throughout the development'. Change to: 'throughout development'

P7 'Stratified by integrated groups, we compared'. Check to see if there's a better word than 'stratified' to use here. As written, it means that 'we', the authors, were arranged into layers by integrated groups.

P7 'WT and mutant cells at the same dominant stages'. Each time that 'dominant' is used, see if 'predominant' would be a better word.

Figure 4A. The triangles are very difficult to see. Would some other visualization work better?

P7 'Figure 4B shows genes that were differentially expressed between WT and mutants.'

Are these all of the genes that were differentially expressed between these two genotypes? Or a subset? And if a subset, then how were these specific genes selected to display?

In addition, be explicit about the comparison. Does red mean that the WT or the mutant is overexpressing? E.g., is *Dmrt1* overexpressed in WT or in mutant?

Figure 4B. The gradient in the main portion of the figure should match the order shown in the scale. The darkest red indeed is on top, but then for the blues, the darkest is on top, not at the bottom as in the scale.

Figure 4c. If id1 and nanos2 are marker genes for GSCs and disappear at later stages in WTs, and if their expression doesn't disappear in mutants, then the mutant cells don't advance to the stage where expression of these genes normally disappear, right? So there wouldn't be any mutant cells at stages G-P to Prog-C(S) because id1 and nanos2 are still expressed in mutant oocytes in these columns in the figure? Or at least these would be variants of the wild type clusters.

In addition, it looks like foxl2l inhibits its own expression at early stages, because without foxl2l activity, the expression of foxl2l is much higher in mutants and WTs in GSCI to Prog-E columns.

P7 'The expression of *Dmrt1* and dele1 in WT was steady throughout the developmental stages (Figure 4C), but *Dmrt1* was abnormally high and dele1 was abnormally low in both mutant GSCI and Prog-C/Prog-C(S).'

The fact that *Dmrt1* was abnormally high in mutants suggests that a role of foxl2l activity is to inhibit expression of *Dmrt1*.

The fact that dele1 was low in mutant Prog-C(S) cells in mutants, suggests that foxl2l activity encourages expression of dele1.

Those conclusions, if authors agree with them, might have more impact here than what's written: 'These results indicate multiple defects of gene expression in the mutants.'

P7 'Co-expression of foxl2l with *Dmrt1*, nanos2 or id1 in foxl2l mutant'. Consider changing to '…in foxl2l mutants', or to '…in foxl2l mutant germ cells'.

Figure 5A legend. Say explicitly that cells expressing both genes are represented by yellow. It's obvious of course from the scale grid but would help some readers to say it explicitly.

P7 'increased in the mutant (Figure 5B). Moreover, the number of nanos2+ cell'

Plural: …of nanos2+ cells…'

P7 '(Figure 5C). To further investigate whether GSC is affected in mutants' '…whether GSCs are affected…'

P7 'counted the number of nanos2+foxl2l- cells, and found that the number increased dramatically in mutants at 26 dpf'. increased dramatically in mutants compared to WTs? Or increased in mutants at 26 dpf compared to some other time in development?

P7 'GSC cells in mutant together'. Plural: 'GSC cells in mutants together'

P7 'GSC cells in mutant together suggest that some Prog-C may revert to GSC in', What is the evidence that they develop to Prog-C and revert rather than that they never made it fully to Prog-C and simply maintained some of the gene expression properties of the earlier stages? These are two competing hypotheses, and both should be mentioned.

Figure 6 legend. Mention 'G-P transition' in the legend or remove it from the sketch. The word 'Proliferation' in black on a dark gray field is hard to read.

P9 'they undergo G to P transition for further differentiation into progenitors.'

Define 'G-to-P'.

P8 'Progenitors include three types: Prog-E, Prog-C, and Prog-L.' It would be good if the text here could state the transcriptomic features that distinguish these three phases from each other like the text did for the two kinds of GSCs.

P9 'The fate of germ cells is determined following the instruction of the supporting cells.' Compare to germ cell sex determination in mammals.

P9 'deficiency, however, leads to cell arrest at Prog-C as shown in this article.'. 'deficiency, however, leads to cell arrest at Prog-C as shown by results presented here.

P9 'commitment to the female fate as shown here. It indicates'. What is the antecedent of 'It'?

P10 'Id1, inhibitor of DNA binding 1, is a transcription regulator that lacks DNA…'.'…that lacks a DNA …'

P10 'stem-cell genes (id1 and nanos2) and male gene *Dmrt1*.' First, '…and the male gene *Dmrt1*'

Second, are there other early male genes that are suppressed by Foxl2l action besides *Dmrt1*?

P10 'The alternative development of foxl2l mutant germ cell'. Plural: '…mutant germ cells'

---

## [Author Response]

Essential Revisions:1) Evidence that committed progenitors revert to germline stem cells (GSCs) is weak. Functional data are required to draw this conclusion. The phrase 'reveal stemness suppression' should be removed from the title and alternative explanations should appear in the text. The claim that suppressing stemness is necessary for female differentiation is not well supported, for example, cells might initiate the committed progenitor program but fail to shut off and continue to express part of the GSC program.

We have removed the term “stemness suppression” from the text and added a discussion of alternative explanations.

2) Essential literature that places the work in context must be discussed, with several cases noted in each specific review.

We have removed the term “stemness suppression” from the text and added a discussion of alternative explanations.

3) Alternative explanations to the statement that committed progenitor stage is "the gate toward female determination" must be discussed, for example, that the late progenitor stage is the key stage for female determination.

We agree that the existence of Prog-L is required for female differentiation. Our results show that *foxl2l* drives the differentiation of Prog-C to Prog-L, and therefore guides progenitor toward female pathway. We have clarified our statement about the importance of Prog-L in female differentiation.

4) Consider alternatives to the conclusion that the progenitors are differentiated because it is based solely on the expression of foxl2l, which is initially expressed in the juvenile ovary state that lab strains develop through.

We do not think that all progenitors are sexually differentiated. Prog-E and GSC probably do not have a sexual identity. Only Prog-L is stepping towards female meiosis*.* Prog-C differentiate into Prog-L, and commits germ cells into the female pathway. We derived this conclusion not only based on *Foxl2* expression, but also based on our whole-genome transcriptomic analysis and *foxl2l* mutant phenotype. Furthermore, *foxl2l* is not “initially expressed in the juvenile ovary state that lab strains develop through”. We have examined the expression of *foxl2l* in the wild Nadia strain and detected the same pattern as that in the lab strains during the larval stage (Figure 2). Our result showed that at 26 dpf, *foxl2l* is not expressed in WT meiotic oocytes, instead it is expressed in two type of progenitor, Prog-C/Prog-C(S) and Prog-L, and its level is reduced after meiotic entry. Our data show that Foxl2l drives the differentiation of Prog-C to ProgL, and the existence of Prog-L is required for female differentiation.

5). Consider alternative explanations for the differences in the action of Foxl2(Fox3) in zebrafish and medaka.

Medaka Foxl2l promotes meiotic entry and the expression of meiotic genes. Zebrafish Foxl2l also has the same functions as shown by our previous RNAseq result (*Dev Biol*, *517*, 91-99, 2024) and the expression of zebrafish *foxl2l* in Prog-L (Figure 1D). Additionally, zebrafish *foxl2l* deficiency leads to activation of *nanos2* in mutated Prog-C. It is unknown whether *nanos2* is activated in progenitor cells of XX *foxl2l^-/-^* medaka. Furthermore, unlike the absence of meiotic germ cells in mutant zebrafish, in *foxl2l^-/-^* medaka, meiotic germ cells are still present in XX juvenile gonads. These differences raise two possibilities. First, Foxl2l in medaka and zebrafish share a similar role in regulating progenitor differentiation. The difference in mutant phenotypes can be due to a stronger female tendency in medaka that prevents complete blockage of female differentiation and meiosis in *foxl2l* mutants. Second, zebrafish Foxl2l has an additional role in driving progenitor differentiation. We have now included the discussion of the alternative explanations about the difference in the action of Foxl2l in these two species.

6) Methods require more details on how fish were selected for scRNA-seq.

We have added more details about the collection of fish used for scRNAseq in the text.

Reviewer #1 (Recommendations for the authors):Specific points.1) Page 7 lines 258-261. The authors conclude that the committed progenitors revert to GSCs based on the coexpression of nanos2 and foxl2l nanos2 and later on the same page line 262, based on expression of id1 in mutants but not WT. Without functional data demonstrating that the progenitors revert to an early state, alternative interpretations should be considered. For example, can the authors exclude the alternative possibility that the cells initiate the committed progenitor program but do not shut off and continue to express the GSC program such that the coexpression of both programs blocks differentiation? It would help to discuss these findings in the context of Fox family functions. Some Fox family members, Foxl2 and FoxPs for example, are known to be both activators and repressors of transcription or act primarily as repressors. Relevant to this work, repressive activity of Foxl2 has previously been reported in the mammalian ovary (Pisarska et al. Endocrinology 2004, Pisarska Am J. Phys Endo. Metabolism 2010, Kuo Reproduction 2012, Kuo Endocrinology 2011, as well as several more recent publications). In this context interfering with Foxl2 repression was proposed to accelerate follicle recruitment and cause premature ovarian failure.

Thanks for your insightful comment. Alternative interpretations are indeed needed. We have examined the expression of GSC markers including *nanos2* and found that in the mutants, *nanos2* or other GSC markers were not significantly upregulated in the GSC-to progenitor transition (G-P) and early progenitors (Prog-E) (Figure 4B). The expression of these GSC markers was also low in the integrated clusters when G-P and Prog-E stages were prominent (Figure 3D and Figure 3E). Only one GSC marker, *nanos2*, was high in mutant Prog-C. These results indicate that perhaps some mutant Prog-C acquires some GSC properties with the upregulation of *nanos2* instead of a continuous GSC program. We have now included both interpretations and clarified our rationale about the mutant cells gaining a new GSC property. We have also included the discussion about the dual regulation of *Foxl2* in the **discussion**.

2) Discussion Page 9 lines 289-293. This section is not entirely clear. First GSC-I and GSC-II subtypes are introduced, then it is stated that "all GSCs self -renew" and a Saito 2007 paper is cited. This is followed by a description of "two types of daughter cells derived from GSC self-renewal". Does the conclusion that "all GSCs renew" still hold in light of the two GSC populations? Would a division that produces a daughter that cannot renew be considered "self-renewal" or instead a non-renewing division?

Our original statement about the self-renewal in GSC was probably not accurate. Here we speculate that the two subtypes of GSC might represent two different daughter cells generated by asymmetric division of GSC. We have revised our discussion in the text.

3) Discussion Page 9 Line 295-7. The authors conclude that the committed progenitor stage is "the gate toward female determination" and that they "stay at S-Phase temporarily before differentiation". Is this conclusion based solely on RNA expression? Also, in several species, including zebrafish, meiotic entry has been correlated with ovary development. Can the possibility that the late progenitor stage, the stage when meiotic genes are detected and a stage missing in foxl2l mutants, is the key stage for female determination be excluded?4) The same comment as above pertains to Page 9 lines 281-284 of the discussion.

We indicate that Prog-C “stay at S-pase temporarily” because we detected a group of Prog-C cells, (Prog-C(S)), that are rich in S-phase genes and nucleosome assembly (*zgc:153405*, *si:ch211-113a14.24*). We refer Prog-C as a gate before female differentiation because the *foxl2l* mutant cells are halted at Prog-C as shown by in situ hybridization, cell counting, and scRNAseq analyses. Prog-L as well as the meiotic stages are completely absent. We agree that the existence of Prog-L is important for the female differentiation, but the changes already start at Prog-C. We have revised our statement to make it more precise.

5) Page 3 Line 64 states that "the developmental stages of germ cells in zebrafish remain unclear". This is somewhat misleading as recently published single cell datasets have analyzed and defined the stages of germ cell development, including during and after sexual differentiation. This should be clarified, and the relevant published work should be cited. (Liu et al. eLife 2022, Wilson et al. Front Cell Dev Bio 2024.)

Thanks for your suggestion. We have cited the references and revised our statement.

6) Page 3 Line 70 might be confusing because it states that "XX medaka fish with foxl3 mutation become hermaphroditic processing functional sperms". Please clarify whether they also produce oocytes or are genetically female "XX" but make sperm.

XX medaka fish with *foxl3* mutation produce both functional sperms and oocytes. Thanks for your suggestion. We have clarified this point.

7) On Page 9 lines 305-307, the authors discuss prior working showing that loss of germ cells leads to male development and that germ cells are required for female development. Then lines 307-309 state that this work shows some progenitors are already sexually differentiated. First, the stages compared are completely different. The early work looks at the primordial germ cells and their loss in the first few days of development before a gonad forms and this work looks after a gonad has formed and when sex is being determined.

Both previous studies and our study indicate the important role of germ cells in zebrafish sex differentiation during gonadal development. The earlier works show that the abundance of primordial germ cells contributes to sex differentiation. Our current finding further suggests the existence of female identify in some germ cells at the juvenile stage and discusses the importance of germ cell in sexual differentiation.

The second concern is that the conclusion that the cells are differentiated is based on the expression of foxl2l, which is expressed in the juvenile ovary state that lab strains have been shown to develop through (Wilson et al. Front Cell Dev Bio 2024). While it is fair to state that some cells express ovary markers, are they really differentiated?

We indicated that the Prog-L cells are differentiated based on the analysis of whole-transcriptomes, not simply by *foxl2l* expression. This analysis is furthermore backed up by functional analysis. In the absence of functional Foxl2l, Prog-L is absent and the mutant gonads eventually undergo male development. Therefore, we infer that Prog-L stage is required for female differentiation and Foxl2l is essential for the differentiation of Prog-C to Prog-L. On a different note about the expression of *foxl2l*, our in situ hybridization data show that *foxl2l* has the same expression pattern in the wild Nadia strain as in laboratory strains at the larval stage (Figure 2). Our analysis of 26-pdf scRNAseq further showed that *foxl2l* is expressed in two types of progenitors and becomes barely detected in the meiotic cell.

For example, in the context of the foxl2l mutant, GSCs and early progenitors inappropriately express foxl2l, but the mutants develop as males. Please clarify.

In *foxl2l* mutants, the mutant *foxl2l* mRNA is stable, but fails to produce functional protein, resulting in the impairment of female differentiation and progenitor development. We detected mutated *foxl2l* transcripts in mutant cells, but this does not mean that these cells contain Foxl2l. These mutants develop into males eventually.

8) Page 9 lines 315-318 compares medaka and zebrafish foxl2l mutants and seems to suggest that Foxl2l is required for meiosis in medaka but has a different role in zebrafish. Is it possible that foxl2l plays a similar role in repressing the earlier developmental programs of GSCs and early progenitors in both cases?

We agree that zebrafish and medaka Foxl2l share the same role to promote female meiosis. But we are not certain whether medaka Foxl2l represses early progenitor development. In *foxl2l^-/-^* zebrafish, meiotic germ cells are absent, while in *foxl2l^-/-^* medaka, XX gonads contain meiotic germ cells. The difference raises two possibilities. First, Foxl2l in medaka shares similar roles in progenitor differentiation and repressing GSC program mekada and as in zebrafish. The different phenotypes can be due to the stronger female tendency contributed by genomic background in medaka that prevents complete blockage of female differentiation and meiosis in *foxl2l* mutants. It would be interesting to know whether the GSC program is activated in progenitor cells of XX *foxl2l^-/-^* medaka. Second, zebrafish has additional role in driving progenitor differentiation. We have now included a discussion of the different phenotypes in these two species.

Is it possible that continued expression of these early programs interferes with activation of meiotic genes? Might this account for the absence of the late progenitor stage in foxl2l mutants since it is defined by expression of foxl2l and meiotic genes?

In zebrafish *foxl2l* mutants, early programs are not continuously activated. GSC marker gene expression is low at the G-P transition and Prog-E stages of *foxl2l* mutants (Figure 3D and 3E), indicating that in *foxl2l^-/-^* mutants, *nanos2* reappears rather than continuous expresses from GSC to progenitor stages. In *foxl2l* mutants, late progenitors and the meiotic stages were completely lost. The loss of late progenitors in zebrafish *foxl2l* mutant is demonstrated by the analysis of the whole transcriptomes rather than merely *foxl2l* and meiotic genes only. Therefore, the absent of meiotic genes in *foxl2l* mutants might be due to the completely loss of late progenitors as well as meiotic cells. This is comparable with differential gene expression observed in Prog-C. The dysregulation of these genes can contribute to the block of Prog-C differentiation resulting in the loss of Prog-L. Moreover, we cannot rule out the possibility that *nanos2* in *foxl2l* mutants might interfere with the activation of meiotic genes as well as other unknown genes. Therefore, it is possible that the absence of Prog-L in mutants is due to an interference of meiotic expression by the differentially expressed genes observed in mutant Prog-C.

9) Page 10 lines 328-330. Direct differentiation of testis without going through a female phase is discussed. Recent work showing direct differentiation in wild strains should also be discussed and cited (Wilson et al. Front Cell Dev Bio 2024).

Thanks for your suggestion. We have added the description and cited this paper.

10) Page 10 lines 359-360. The authors state "Foxl2l may ensure female differentiation by preventing stemness and antagonizing male development." It is unclear why suppressing stemness would be necessary for female differentiation since female zebrafish have stem cells as do male zebrafish. It seems likely that turning off the GSC and early differentiation programs is important for allowing expression of meiosis and oocyte differentiation genes.

development. It is true that we have not proved whether suppression of stemness is required for female differentiation. Yet as this reviewer pointed out, it is likely that turning off the GSC and early differentiation programs is important for allowing expression of genes in meiosis and oocyte differentiation, thus leading to final female differentiation. To avoid confusion, we revised the discussion and speculation about the possible actions of Foxl2l in the text.

11) Page 11 370-374. "Thus, p53 may assist alternative differentiation of mutant germ cells". What is the evidence that p53 is involved in differentiation of germ cells? Sex bias has not been associated with the published p53 mutants. Is it possible that p53 eliminates germ cells that are simultaneously expressing programs of multiple stages? While p53 has been shown to be important for ovary to testis transformation in mutant contexts in adults, it is dispensable in mutants that disrupt ovary development at earlier stages. Please discuss in the context of, and cite, the relevant ovary literature Rodriguez-Mari et al. PLoS Gen 2010, Shive PNAS 2010, Hartung et al. Mol. Reprod. Dev 2014, Miao Development 2017, Kaufman et al. PLoSGen 2018, Bertho et al. Development 2021.

Thanks for your comments and suggestion. The *tp53^-/-^foxl2l^-/-^* double mutant cannot alleviate the all-male phenotype of *foxl2l^-/-^* mutant (*Dev Biol*, *517*, 91-99, 2024), indicating that the male development is not due to p53-mediated germ cell apoptosis. We have compared the relation of *tp53* between these mutants and cited the suggested papers. The involvement of *tp53* in germ cell differentiation is our assumption derived from high *tp53* expression in *foxl2l* mutants in our scRNAseq result. Since p53 was enriched in certain mutant clusters, and the *tp53* mutation fails to rescue the all-male phenotype, it is possible that *tp53* expression in these mutant clusters may play roles other than inducing apoptosis. Although unknown in germ cell differentiation, p53 has a role in promoting the differentiation of airway epithelial progenitors and embryonic stem cells. We have pointed out in the **Discussion** that the notion about the role of p53 in germ cell differentiation is our assumption.

12) Figure 2D. It would be helpful to show the individual fish points or to present the data as a violin plot.

Thanks for your suggestion. We have included the individual fish points in Figure 2D and revised the figure legend.

13) Figure 2F. The numbers are low here only 3-4 gonads are examined for each genotype and marker and in the mutants only 1/4 expresses the "testis" markers.

We performed this staining experiment at 28 dpf. Since 28 dpf is the beginning of male development, it is expected that male morphology only starts to appear in a portion of the gonad. The number does not need to be high. The presence of the testis marker gene expression already indicates male differentiation.

14) Figure 6 model. What is the evidence that prog C becomes GSCs again/revert versus continuing to express earlier programs?

Thanks for your comments. We have examined the expression profile of GSC markers including *nanos2*, *cbx7a*, *psmb13a* and *dnmt3bb.1* in mutant cells. All of them showed low expression in GSC-to-progenitor transition (G-P) stage and Prog-E stage in mutants (Figure 3D, 3E). In the DEG analysis, no GSC markers were upregulated in GSC-to-progenitor transition (G-P) or Prog-E. Only *nanos2* were abruptly upregulated in Prog-C. This result indicates that the GSC program was shut off during the differentiation from GSC to progenitor. Therefore, it is more likely that in the absence of functional Foxl2l, Prog-C acquires some GSC property with upregulation of *nanos2*.

Reviewer #2 (Recommendations for the authors):The evidence that foxl2l cells are arrested at the committed progenitor stage (a possible novel phenotype) is based on the absence of expression of rec8a and Sycp3. Another explanation is that these gonads have already differentiated as testes.

The arrest of committed progenitor stage in mutant is based on the whole transcriptomic profile (Figure 3A), rather than merely the expression of *rec8a* and *sycp3*. The *foxl2l* mutant germ cells failed to be classified as Prog-L or meiotic stages. Consistently, marker genes for Prog-L and meiotic stages including *sycp3* and *rec8a* were consistently at low levels (Figure 3—Figure Supplement 1) (Figure 3B). The *foxl2l^-/-^* gonads will eventually become testes and their germ cells will be differentiated as sperm cells, but not at this early stage. At 26 dpf when scRNAseq was performed, we could not detect the expression of male genes including *tekt1* and *odf3b* in our analysis. Therefore, the mutant cells were not yet differentiated into testis at 26 dpf.

Is rec8a also expressed in male germ cells or is it female germ cell specific? Germ cells in testes are usually delayed in entering meiosis (and thus expressing Sycp3) relative to females so it is not clear if this represents the expected trajectory of testis vs. ovary or a novel developmental state caused by absence of foxl2l.

The *rec8a* transcript can be detected in both testes and ovaries as shown before (DOI: 10.1371/journal.pgen.1009127). The male characteristics were detected in mutant gonads at a later stage, 28 dpf. Some mutant Prog-C cells additionally acquired a novel developmental state (*nanos2^+^*) caused by the absence of Foxl2l. Therefore, it is not a simple testis development from indifferent gonads, rather, the mutant germ cells went through a process involving the untimely GSC gene expression at 26 dpf.

Was expression of any male-specific genes identified in the mutant data set (e.g. tekt1 or odf3b) which could indicate these germ cells were already committed to spermatogenesis?

We did not detect transcripts of male markers including *tekt1* and *odf3b* in mutant cells at 26 dpf by scRNAseq. Therefore, these mutant cells probably have not entered spermatogenesis at this stage.

Also, by "arrested" do the authors mean that they are in a cell cycle arrest? If so, this can be assayed with BrdU or EdU labeling.

The word “arrested” here means “blocked”. Cells cannot be further differentiated to the next step (Prog-L). It does not involve cell cycle arrest. We have revised the sentence to indicate the arrest of germ cell development rather than the arrest of cell cycles in the mutants.

L99: incomplete sentence.L128: The publication associated with this dataset should be referenced (it's referenced in the figure legend but should also be referenced in main text).L138: Using a negative result (absence of staining) to argue for a particular cell stage (prog-E) is problematic. To bolster this argument the authors should consider pointing out that this result is consistent with the scRNA-seq data that predicts little to no overlap between nanos2 and foxl2l expressing cells. Perhaps adding the nanos2 and foxl2l and rec8a UMAPs to Figure 1 would help.

Thanks for your suggestion. We have done the revision.

L148 and Figure 2A. Wilson et al. (2024) showed that the gonads of Nadia ZZ fish appear to directly develop into testis without producing oocytes during the bipotential stage. The data in Figure 2A showing foxl2l expression in the ZZ gonads is therefore unexpected. Can the authors comment on this?

The expression of *foxl2l* is dynamic during development. At the larval stage, *foxl2l* is expressed in all bipotential gonads in both ZZ and ZW. However, during the sexual differentiation stage, *foxl2l* expression gradually is decreased in the male gonads while sustained in the female gonad (ZW) at juvenile age. Many sex-specific genes are first expressed in bipotential gonads before their expression becomes sex-specific; these genes include *Sox9* (*Development* 137:3921-3930, 2010), *Dmrt1* (*Nature* 402:601-602, 1999), and zebrafish *cyp19a1a* (aromatase, *Dev Biol,* 344:849-856, 2010). Therefore, *foxl2l* is not the only gene that is expressed in early bipotential gonads, irrespective of the future sex identity of the gonad. The functional relevance of this early expression in all indifferent gonads has not been investigated.

L154: "…barley observed in testis." Should this be "…not observed…", or is there low but detectable expression of foxl2l in the testis (not clear from Figure 1D)?

Thanks for your suggestion. The expression of *foxl2l* was not observed in the testis by ISH. We have done the revision.

L170: It is not clear from the Methods how the fish were selected for scRNA-seq. With the piwil1:egfp line it is possible to distinguish wild-type males from females at the 27 dpf timepoint used in this analysis based on the size of the gonad (large=female, small=male). Were only fish with large gonads used for WT and foxl2l mutant scRNA-seq? If fish were chosen randomly, it would be expected that germ cells from testes would also be included.

The sizes of male versus female gonads can be distinguished only when these gonads are carefully examined under a fluorescent microscope. To assure the quality and abundance of active germ cells for RNA isolation, one has to dissect fish trunks quickly without checking gonad size. Therefore, germ cells from both presumptive male and presumptive female gonads were included. This is why we did not find obvious male developmental trajectory from our WT database. In zebrafish, around ~80% of juveniles undergo female development first with oocyte development (*Tissue and Cell* 72:101545, 2021). Only ~ 20% of juveniles are presumptive male remaining as undifferentiated germ cell (gonocyte) before male development after 28 dpf. Our scRNAsq was done in germ cells of 26-dpf gonads, earlier than the initiation of typical male development after 28 dpf. Therefore, germ cells in these 20% presumptive male gonads were undifferentiated GSC or progenitor type, which may be quite similar to those cells from presumptive female gonads. Thus, it is reasonable that we did not detect male cell clusters in the WT scRNAseq dataset.

When do foxl2l mutant gonad first start looking like testes? Is there a delay in testis development relative to wild-type?

Spermatogenesis started to be observed in some *foxl2l* mutant gonads at 28 dpf, as shown by DAPI staining and the staining of male markers, *tekt1* and *odf3b* (Figure 2F). This time point of male development is not delayed compared with that in WT.

Figure 2B: Including age-matched wild-type testes in Figure 2B may help clarify if the foxl2l gonads are different or similar to a normal male gonad at this stage of development.

We have published the histology analysis of mutant testis and showed that mutant testis has the same histology as the WT testis. Here we have cited this paper.

Figure 2D: I would suggest replacing si:ch211-191j22.7 with meiosin, since this is the clear ortholog.

Thanks for your suggestion. We have done the revisions.

Figure 3B: It is hard to see the morphology of the nuclei in these images. Grey scale is perfect for the DAPI channel, but it would be helpful to increase the apparent signal intensity using an image processing program like Photoshop so that it is possible to better assess the morphology of nuclei.

Thanks for your suggestion, we have revised figure 3B as suggested.

Figure 3C: I am having a hard time understanding the meaning of Figure 3C. For the WT cells, the nanos2+ cells are restricted to the W1 subcluster (Figure 1D), indicating that the GSCs are homogeneous with respect to gene expression. However, when WT cells are clustered with mutant cells the WT GSCs now partition to two different subclusters (I1 and I2). When the "control" cells are not clustering tightly, it calls into question how much this clustering represents reality. How do the authors rationalize this?

During clustering analysis, its granularity is influenced by the dataset size and is adjusted using a resolution parameter. To ensure robustness, we selected the optimal resolution by identifying the value that produced the highest average silhouette score within a feasible range. This led to the clustering of all GSCs into a single group in the WT dataset. When WT cells were co-clustered with mutant samples, the increased number of GSCs enabled higher-resolution clustering, leading to the identification of two distinct clusters, GSCI and GSCII. These clusters displayed varying levels of upregulation in marker genes: one subset of WT GSCs exhibited higher *nanos2* and lower *dnmt3bb*.1 expression, while the other subset showed the reverse pattern (Figure 1—Figure Supplement 3). While this subdivision is more apparent in the co-cluster dataset, it is not unique to mutants. In both WT and co-cluster analyses, subsets of WT GSCs displayed distinct transcriptomic profiles, in which *nanos2* and *dnmt3bb*.*1* were high in different subsets of GSC cells (Figure 1—Figure Supplement 3). The high expression of *nanos2* and *dnmt3bb*.*1* in different subsets of WT GSCs can also be observed in 40-dpf ovaries in a public database (DOI: 10.7554/*eLife*.76014) shown below. Therefore, these findings support the presence of GSC subtypes in both WT and mutant samples.

L194: Figure S7B is unfortunately not helping me understand how the data was processed. What is meant by "transfer developmental stage?" Does this require/assume that the mutant cells be a specific stage that can be mapped directly onto wildtype stages? I wonder if forcing the mutant cells to cluster with wildtype cells is somehow skewing the results. How different is the clustering if only mutant cells are used?

We used the term "transfer developmental stage" directly from the name of the software *TransferData* in Seurat v3.2.3. This process aligns mutant cells with developmental stages previously defined in WT cells, based on transcriptomic similarities as described in the **Materials and methods**. The function *TransferData* in Seurat v3.2.3 allowed the transfer of the developmental stages from the WT cells to the mutant cells. Initially, principal components (PCs) were derived from the WT cells based on the expression of their cluster markers. Mutant cells were then projected to the same PCs. The PCs and the projected PCs were used to identify the mutual nearest neighbors across mutant cells and WT cells. These mutual nearest neighbors were considered anchors and scored based on the consistency across the neighborhood structure of each dataset. Lastly, the developmental stage of each mutant cell was determined using a weighted voting mechanism. This process considers the stages of WT cells in every anchor, with votes weighted according to the anchor score and the proximity between the mutant cell and its corresponding anchor mutant cell. This approach assumes that mutant cells can indeed be mapped to specific WT stages through their transcriptomic profiles.

To mitigate potential bias, we subsequently conducted a co-clustering analysis of WT and mutant cells, employing a data-driven methodology to identify both shared and unique germ cell subtypes. This co-clustering does not rigidly assign mutant cells to predefined WT stages; rather, it delineates the transcriptomic relationships between WT and mutant cells, facilitating the identification of potentially novel mutant subpopulations.

To further address your concern, we performed an independent clustering analysis using only mutant cells as suggested. The results were highly consistent with those from the co-clustering analysis (see the confusion matrix below; Adjusted Rand Index: 0.8). This independent analysis reinforces the robustness of our conclusion.

In response to your feedback, we have revised the term "transfer developmental stage" to "infer developmental stage" and updated Figure 3—Figure Supplement 2 to include detailed descriptions of each step for greater clarity.

**Author response image 1. sa2fig1:** 

Figure 3E: nanos2 and cbx7a are listed twice. It is likely they should be psmb13a and dnmt3bb.1, respectively.

Thank you for your insightful question. Indeed, the labels of two genes (*nanos2* and *cbx7a*) were mistakenly duplicated in Figure 3E. We have replaced them with the correct gene names: *pasmb13a* and *dnmt3bb.1*.

Was expression of any male-specific gene identified in the mutant data set (e.g. tekt1 or odf3b) which could indicate these germ cells were already committed to spermatogenesis?

We examined the expression profiles of the male-specific genes *tekt1* and *odf3b* within our dataset. In WT cells, both genes exhibited minimal expression prior to the onset of meiotic prophase. Specifically, *tekt1* was virtually undetectable, while *odf3b* was expressed in only a limited subset of WT cells during the GSC and G-P transition stages [(refer to Author response image 1)].

In the mutant dataset, the expression levels of *tekt1* and *odf3b* similarly remained low and were restricted to a small fraction of cells. Importantly, no statistically significant upregulation was observed when compared to their WT counterparts within the same co-clustered groups [(refer to Author response image 2)]. We, therefore, conclude that at 26 dpf, no sign of spermatogenesis can be detected in these germ cells.

**Author response image 3. sa2fig3:** 

Sup Figure S8A: It would be helpful if this figure also included the UMAP shown in Figure 3C for reference (with I1-I7 labels). S8A is hard to interpret given the altered gene expression pattern of mutant cells. Some genes fit the proposed cell stages (e.g. dnmt3bb.1), but others are not as clear (e.g. nanos2, dhrs1).

We are not sure about this reviewer’s point about supplementary Figure S8A. Our original supplementary Figure S8A shows violin plots of differentially expressed genes but does not show UMAP and does not contain *dnmt3bb.1* and *dhrs1*genes. Thus, supplementary Figure S8A is unrelated to the UMAP in Figure 3C. We guess this reviewer may want to comment on the UMAP of the original supplementary Figure S6A. To improve clarity, we have reorganized the supplementary figures and added our analysis methods to enhance readers’ comprehension. The UMAP in the revised Figure 3**—**Figure Supplement 1C (original Figure S6A) shows the inferred developmental stage of mutant cells using the method shown in the revised. Figure 3**—**Figure Supplement 1B (original Figure S6C) showed that marker of early stage (GSC, G-P, Prog-E and Prog-C/Prog-C(S)) were upregulated in distinct subset of mutant cells. For example, *nanos2* is highly expressed in a different subset from *dhrs1*. The other analysis is co-clustering analysis. The flow chart is shown in the new Figure 3**—**Figure Supplement 2B and the co-cluster result is shown in Figure 3C.

L210: "This suggests that GSCs can be further divided into two finer stages, GSCI and GSCII." If you only see this subdivision in mutants, then it is hard to argue that these are representative stages present in WT.

We acknowledge your concern regarding the subdivision of GSCs into GSCI and GSCII. While this subdivision is more apparent in the co-cluster dataset, it is not unique to mutants. In both WT and co-cluster analyses, subsets of WT GSCs displayed distinct transcriptomic profiles, in which *nanos2* and *dnmt3bb*.*1* were high in different subsets of GSC cells (Figure 1—Figure Supplement 3). The high expression of *nanos2* and *dnmt3bb*.*1* in different subsets of WT GSCs can also be observed in 40-dpf ovaries in a public database (DOI: 10.7554/*eLife*.76014) Author response image 3. These findings indicate that GSCI and GSCII subtypes exist in both WT and mutant samples, reflecting shared biological heterogeneity.

**Author response image 4. sa2fig4:** 

Line 307: It is not clear what this statement means. It appears to imply that foxl2l-expressing cells are sexually differentiated. However, in Figure 2A it is shown that foxl2l is expressed in germ cells of ZZ fish during the bipotential stage, yet these germ cell presumable would have differentiated as sperm, not oocytes.

Thanks for your comment. Foxl2l might play diverse roles during the bipotential stage and sex differentiation stage. In bipotential gonads, *foxl2l* is expressed in both female (ZW) and male (ZZ) gonad, although it is considered a female-specific gene (Figure 2A). Many sex-specific genes are initially expressed in bipotential gonads before they become sex-specific. For example, mouse *Sox9* is expressed in both XX and XY gonads before testis differentiation (Nat Genet, 14:62-68, 1996). The same for *Dmrt1* from mouse, chicken, and alligator (Nature, 402:601-602,1999)*.* The roles of these genes in the initial indifferent gonads are still unknown at the moment.

L302-322: The authors need to spell out the evidence that supports these statements. It could be argued that the difference is not in the role of foxl2l between Medaka and zebrafish, but the fact that medaka has a strong genetic sex determination mechanism while zebrafish does not. Thus, the apparent difference in phenotype is because medaka cannot sex reverse as easily as zebrafish.

Thanks for your comment. We have toned down our speculation. We pointed out the difference in phenotypes between zebrafish and medaka and the possibilities that cause the difference, including the different roles of Foxl2l between these two species or the existence of sex chromosome in medaka.

L331: For completeness, please add a reference to Wilson et al., (2024; PMID: 38529407)

Thanks for your suggestion, we have done the citation.

L356: The authors argue that one role of Foxl2l is to suppress Dmrt1 expression in progenitor cells, yet in the foxl2l mutant germ cells, Dmrt1 is upregulated in all mutant cell stages, from GSC to prog-C cells (i.e. even in cells that do not normally express foxl2l expression). It is possible that the upregulation in all cells is because the gonads from which these cells were isolated have already begun to differentiate as a testis and the germ cells are now more male-like than female-like. This would also suggest the possibility that male and female GSC's do not express equivalent amounts of Dmrt1.

DEG analysis in Figure 4 showed that the upregulation of *Dmrt1* is significant only in GSCI, Prog-C and Prog-C(S) in foxl2l mutants. We cannot find the upregulation of other male genes in mutant cells. Therefore, it is more likely the upregulation of *Dmrt1* in some mutant Prog-C(S) and GSCI is due to the loss of functional Foxl2l rather than the initiation of the spermatogenic program.

Reviewer #3 (Recommendations for the authors):The major conclusion, i.e., title and short title may be misleading and may cause confusion. Stemness is the ability of a cell to perpetuate/continue/preserve its lineage, to give rise to differentiated cells. With foxl2l, progenitor cells committed cell (Prog-C) will continue to develop into late progenitor (Prog-L). In addition, there is no strong evidence that Prog-C was able to dedifferentiate back into GSC in foxl2l mutants. Prog-C could be simply lost in foxl2l mutant, and increased nanos2 cells could be due to increased proliferation of GSC. Therefore, "Foxl2l suppresses stemness" could be misleading.

Our data show that *nanos2* expression was low in mutant cells at the GSC-to-progenitor transition stage and Prog-E stage, but become upregulated only in Prog-C (Figure 4C), indicating that mutant Prog-C may acquire some GSC property with upregulation of *nanos2*. Moreover, in the mutant gonad, both the *nanos2*-expressing Prog-C and the *foxl2l^-^nanos2^+^* GSC were increased compared with WT. These results raise one possibility that some *nanos2*-expressing mutant Prog-C may revert to GSC in *foxl2l* mutant. Although *foxl2l* is absent in WT GSC, it is also possible that the loss of Foxl2l triggers the proliferation of GSC. We have added this possibility to the **Result.** We agree to tone down our statement and remove the word “stemness suppression” from our title.

Is there size and morphological difference between prog-C and prog-L?

No. The size and the morphology were similar between Prog-C and Prog-L.

Lines 23-24, 64-65 and in other parts of the text "germ cell types … remain elusive" "developmental stages of germ cells in zebrafish remain unclear." Probably should be specified as early germ cell types. Five developmental stages of germ cells (oocytes) in zebrafish are well defined by Kelly Selman since 1993. In fact, this manuscript was focusing on fine classification and separation of different germ cell types prior to stage I, which should be highlighted or emphasized, because this is not possible until scRNAseq became available.

Thanks for your suggestion, we have done the revision.

The label D for figure 1D is too far, should be close to the sub-panel.

Thanks for your suggestion, we have revised figure 1D.

It is hard to view two different symbols illustrated in Figures 3C and 4A, suggest to separate symbols into side-by-side figures.

Thank you for the feedback. We have revised Figures 3C and 4A so that the symbols are easier to read.

The cell numbers for mutant in figure 3D are much higher than those in WT, why?

We do not know the exact reasons why we obtained more mutant germ cells than WT. The initial number of sorted cells for scRNA-seq was comparable between the two groups, with 3,231 WT cells and 4,019 mutant cells collected. Following sequencing, 1,677 WT and 2,947 mutant barcodes were identified as containing intact cells using the default settings of the 10x Cell Ranger pipeline. Subsequently, we employed a rigorous two-step quality control procedure to eliminate low-quality cells, such as damaged cells and doublets, as detailed in the "Quality Control" section. To ensure the robustness of our analysis, we retained 769 high-quality WT cells and 2,399 high-quality mutant cells for downstream analyses. The greater retention of mutant cells after quality control may reflect variations in cellular properties, such as viability or transcriptomic integrity between the WT and mutant samples. It can be random or because of the higher numbers of germ cells at the Prog-C stage in the mutant gonads due to germ cell arrest and accumulation.

Why are there two nanos2 lines (lines 1 & 3) in the same GSC group with different expression profiles in Figure 3E? nanos2 expression was not listed for the mutants in figure 3E. Y-axis had label for mutant, but does not have the label for WT, not was explained in the figure legend.

Thanks for pointing out our typographical error. The labels of two genes (*nanos2* and *cbx7a*) were incorrectly duplicated in Figure 3E. We have replaced them with *pasmb13a* and *dnmt3bb.1*. The original label “mutant” in the Y-axis of Figure 3E is misleading. We have changed to “Mutant Specific Markers”, which indicates the expression of mutant-specific markers in the mutant cells of each integrated cluster (I1-I8). The expression *nanos2* in the mutant cells of each integrated cluster (I1-I8) is shown in line 1.

Lines 171-173, without foxl2l, which is a transcriptional factor, the gene expression would change. So, is it still validate using wildtype marker genes to define developmental stages found in foxl2l.

While we recognize that the absence of Foxl2l may alter the transcriptional landscape in mutant cells and we indeed detected many changes in gene expression as shown in Figure 4, our analysis demonstrates that many WT marker genes remain expressed across distinct subsets of mutant cells (Supplementary Figure S6A). Additionally, we also detected many cells that uniquely expressed mutant-specific marker genes (Figure 3E). These findings support their utility in delineating mutant germ cell types despite the altered gene regulatory environment.

Furthermore, we acknowledge the presence of unique early germ cell types in the mutant population. To address this, we performed a co-clustering analysis of WT and mutant cells based on their comprehensive transcriptomic profiles. This approach does not assume a direct correspondence between mutant and WT developmental stages. Instead, it facilitates the identification of novel germ cell types in mutants while characterizing the relationship between WT and mutant cells. The eight integrated groups (I1-I8) reveal both shared and distinct transcriptomic features, providing deeper insights into the altered developmental program of foxl2l mutants (Figure 3C).

Can different types of early germ cells be separated using scRNAseq data from the foxl2l mutant? Can pseudotime plots be used for the foxl2l mutant? If so, what looks like?

Trajectory and pseudotime analyses depend on critical assumptions regarding the selection of root cells and the underlying structure of cellular trajectories, whether they follow a tree-like, linear, or circular pattern. The inferred trajectories are highly sensitive to parameter choices, which are often informed by biological hypotheses and thus carry an element of subjectivity. Most trajectory inference tools, including Monocle3, are optimized for tree-like structures and may require parameter adjustments to accurately model looped or cyclic trajectories as hypothesized for the mutant cells. A pseudotime plot for the mutant cells may thus appears meaningless. To ensure a rigorous and transparent analysis, we opted not to include potentially subjective trajectory and pseudotime results, presenting only similarity-based analysis in the main text.

Abnormal increased expression of nonfunctional foxl2l (figure 3F) in foxl2l mutant suggests a feedback mechanism.

Yes, I agree that there is a feedback mechanism for *foxl2l* expression although we did not emphasize this point in the text. Figure 3F shows the increased number of *foxl2l^+^* cells in *foxl2l* mutant. We also found that *foxl2l* transcripts were aberrantly upregulated in only mutant Prog-E (Figure 4B). It is possible that the Foxl2l might have feedback regulation in Prog-E. It is also possible progenitor transit from Prog-E to Prog-C might be defective.

Co-expression and increased expression of nanos2/id1 with foxl2l in GSC in foxl2l mutant is interest, as these two genes are expressed separately in WT. Can overexpress of nanos2 and id1 lead to loss of oocytes, or dedifferentiation of oocytes (Prog-C) back to PGCs, and female to male sex reversal?

It is possible that overexpressing *nanos2* or *id1* triggers the dedifferentiation of Prog-C back to PGCs, or prohibits the development of oocytes. However, in the *foxl2l* mutant, the expression of quite a few genes was altered. Therefore, overexpress/knockout only one or two DEGs might not be sufficient to replicate these *foxl2l* mutant phenotypes.

Guest Editor (suggestions for the authors)The Hsu et al. manuscript presents valuable data that extend earlier single cell transcriptomic data on zebrafish gonads to help us better understand various initial stages of germ cell development in zebrafish and the role of Foxl2l in oogenesis. Data are well presented. Some of the conclusions do not seem to be well supported by the data, or alternatives are sometimes not considered.The authors investigated development of zebrafish germ cells by isolating Piwil1-expressing cells and performing scRNA-seq in wild types at 26 days post fertilization, which augments published single cell transcriptomic studies of whole zebrafish gonads at 40 dpf, and they performed scRNA-seq experiments on mutants lacking foxl2l activity. These data, validated by in situ hybridization studies, allowed the identification of new stages of germ cell development. Studies showed that foxl2l mutants failed to develop meiotic oocytes, and that mutant germ cells reached a modified committed progenitor stage (Prog-C) but did not become late progenitors (Prog-L). The cells the authors called Prog-C cells in mutants, unlike wild types, expressed some GSC markers, suggesting that they maybe shouldn't be called Prog-C cells in mutants. From this result, the text concluded that committed progenitors then reverted to germ cell stem cells (GSC), which then directly developed into spermatogenic cells, and hence the animals became males, concluding that this was a direct development into males.Alternative hypotheses exist for these data. The expression of genes normally expressed by GSCs (e.g., nanos2) in what otherwise seem to be committed progenitors was interpreted as a reversion of Prog-Cs to GSCs. An equally possible hypothesis is that the normally GSC-expressed genes never turned off and that the mutant 'Prog-C' never fully developed. Both explanations need to be presented and arguments assessed for and against each of these explanations.

Thanks for your comments. We have contemplated alternative explanations for our results. Regarding the possibility of continued GSC programs in the mutants, we examined the expression profile of GSC markers including *nanos2*, *cbx7a*, *psmb13a* and *dnmt3bb.1* in mutant cells. All of them showed low expression in GSC-to-progenitor transition (G-P) stage and Prog-E stage in the mutant (Figure 3D, 3E). In DEG analysis, no GSC markers were upregulated in GSC-to-progenitor transition (G-P) or Prog-E. In Prog-C, however, *nanos2* was abruptly upregulated. This result indicates that the GSC program was shut off during the differentiation from GSC to progenitor. It was re-expressed in the mutant at the Prog-C stage. Therefore, it is more likely that in the absence of functional Foxl2l, Prog-C acquires some GSC property with upregulation of *nanos2*.

Additionally, the conclusion that there was direct male differentiation in foxl2l mutants without going through the female phase has alternative explanations consistent with the data. This direct male development argument assumes that the Prog-C stage has not already embarked on a female pathway. The text did not demonstrate that Prog-C, or even the earlier stages like Prog-E, had not already begun to be female. The very name 'committed progenitor cells' suggests that these cells are already committed to the female pathway in wild types.

Our transcriptomic analysis showed that both male genes (eg. *tekt1*, *odf3b*) and genes related to female meiosis (e.g., *rec8a*, *meiosin*) were barely detected in Prog-E and Prog-C, suggesting that male or female pathway has not yet been initiated in Prog-E and Prog-C. Furthermore, in Prog-L, the female meiotic genes start to express highly, suggesting that Prog-C is the commitment stage toward either female differentiation or remaining indifferent, followed by male differentiation in the future. This is why the term “committed progenitor” is used.

These authors and others have shown that one of the earliest steps in female gonad development in zebrafish is an increase in the number of primordial germ cells compared to zebrafish that will become males, by 14 days post fertilization. The cell division marker analysis reported here suggests that those PGC proliferation steps are already occurring in the cells studied here. So do the 'Proliferation' arrows shown in Figure 6 for both the Prog-E and Prog-C stages. Thus, these stages would already be female-oriented germ cells. And thus, when the reversion of Prog-C towards Prog-C-to-GSC happens, what that really represents is a reversion of female pathway cells towards uncommitted GSCs that then embark on a male pathway, as happens in many zebrafish individuals due to stress or temperature or other features. Thus, the data presented here shows that the male pathway is taken after germ cells have embarked on the female pathway and turned on foxl2l, but subsequently thwarted from becoming a meiotic oocyte due to the lack of Foxl2l in the mutants.The title (Single cell transcriptomes of zebrafish germline reveal stemness suppression and progenitor feminization by Foxl2l) should be revised because the conclusion that Foxl2l suppresses 'stemness' is not well supported. The data do show that foxl2l mutants have cell types that co-express GSC markers like nanos2 with genes usually expressed later in female germ cell development, like sycp1 and sycp3 (Figure S6), but that does not clearly mean that these mutant cells are in fact stem cells.

Thanks for your insightful comments. Only Prog-L expressed female meiotic genes, indicating these cells already differentiated towards females. The earlier progenitors, Prog-E and Prog-C, are indifferent progenitors because they lack transcripts of both male and female marker genes. Prog-C is the commitment stage deciding either to initiate female differentiation or remain indifferent, waiting for future male differentiation. Here in our mutant study, we found *foxl2l* is critical for the development of Prog-C toward Prog-L. Therefore, in the *foxl2l* mutant, Prog-C fails to develop into Prog-L and fails to undergo female differentiation. Taken together, we suggest that in the *foxl2l* mutant, female differentiation is fully blocked, and some of the Prog-C start to express the GSC marker *nanos2* through an unknown mechanism. We agree to remove the term “suppression of stemness”.

The 'allmarkers' files are not included in the supplementary data, making it impossible to see how genes were selected for display. These files must be added to the supplementary data.

We have now added the files in our supplements.

Some suggestions for improving clarity in the text and figures.P2 'factor, Foxl2l, is expressed in the progenitors committed to the ovary fate.'. Consider saying '…in the germ cell progenitors committed to the oocyte fate..'P2 'Another single-cell profiling of foxl2l-/- germ cells reveals the arrest of early progenitors.'Text here uses both present and past tenses. For describing the results of experiments done in the past, past tense might be better, as the text often uses in 'mediated mutation of foxl2l produced 100% male fish' and 'in stem cells) was elevated'. So maybe use 'revealed' instead of 'reveals'. And again could say '…early germ cell progenitors.' Just to be precise. Likewise, '…of nanos2 (germ cell stem cell)…'.In Key Words, consider changing 'stem cell' to 'germ cell stem cell', and 'germ cell progenitor', which might better direct interested folks to this paper. Consider changing 'female' as a keyword, maybe it's too generic, to 'female-to-male sex change', which might get more focused hits.P2 'increase of nanos2+ germ cell in foxl2l'. Use here plural, 'germ cells'…P3 'contrast to higher vertebrates,'. Consider saying what specific taxon is meant here rather than 'higher'. Talking about higher and lower animals comes from a pre-scientific view of the scale of life.P3 'system, but domesticated laboratory strains have no sex chromosome'. Use plural, 'chromosomes'.P3 'the differentiation of zebrafish gonad. Zebrafish'. Plural, 'gonads'P3 'of germ cells (nr0b1 mutants) or no germ cell (dnd morphans) develop'. Plural, and spelling, 'or no germ cells (dnd morphantsP3 'However, the developmental stages of germ cells in zebrafish remain unclear.'I think the developmental stages have been clear, but the molecular genetics that moves cells through these stages is unclear, and the authors' work subdivides certain stages.P3 'and initiates oogenesis in XX gonad Kikuchi'. Plural, 'XX gonads'P3 'with foxl3 mutation become hermaphroditic possessing'. Several changes: 'with a foxl3 mutation become hermaphrodites possessing functional sperm'P3 'and proved that Foxl2l controls the development of committed progenitor,'. Plural, 'progenitors'.P4 'Germ cells play important roles in zebrafish sex differentiation, but their development is poorly studied.'Maybe more accurate to say: 'Germ cells play important roles in zebrafish sex differentiation, but we still have much to learn about their development.'

Thanks for your suggestion. We have done the revision.

P4 '2002). Fluorescent germ cells were isolated from piwil1:EGFP transgenic fish by cell sorting'The reader needs to know when piwil1 turns on in germ cell development with respect to the timing of oogonia vs. spermatogonia commitment. We need to know if by using the transgene we miss early but important stages in germ cell development.

According to the previous study (doi: 10.1002/dvdy.22404), in *piwil1:EGFP* transgenic fish, EGFP is expressed in all stages of germ cell in gonad at larva and juvenile stages. In ovary, EGFP is highly expressed in oogonia and stage I oocyte. In testis, EGFP can be detected in all stage of spermatogenic cells. Therefore, to obtain all germ cells for analysis, we chose *piwil1:EGFP* transgenic fish for germ cell sorting. We have now described the cells that can be labeled by piwil1:EGFP in the text.

Supplementary Figure S2 must have a mechanism for the reader to identify the genes in the rows. They would be too small to read if written on the figure so it should be in an Excel file in the supplement.

Thank you for the suggestion. To enhance accessibility and readability, we have included a detailed list of the genes represented in Figure 1—Figure Supplement 2 as an Excel file, titled ["Supplementary_File_Genes_FigureS2.xlsx"], within the supplementary materials. This provides an efficient means for readers to access the information without compromising the clarity of the figure.

P4. 'expression profiles of the top 37 out of a total of 1491 marker genes'Top based on what criterion? P-adj? P? Log2 fold change? Most specific 2 or 3 per cluster? Or something else? And what defines a 'marker gene'?

Thanks for your suggestion. For each cluster, genes that satisfy the following criteria: (i) specificity ≥ 0.2, (ii) Q-value < 0.01, and (iii) having non-zero counts in ≥ 80% of cells in the cluster were selected as markers. Marker genes were ranked based on the specificity within clusters, and the top 2 to 4 genes in each cluster were selected as top markers. We have revised the **Result** and defined criteria of marker genes and top marker genes in the “Cluster marker identification” section within “**Materials and methods**”

P4 'markers (sycp1, sycp3, dmc1, buc and zp3e). Are buc, which helps assemble germplasm, and zp3e, an eggshell component, \involved in meiosis? Consider rephrasing.

Thanks for your suggestion. We have done the revision.

Suppl Figure 3 and 4. Although most of the markers show very similar patterns, thus cross-validating both datasets, it looks like for the markers CU467646.3, si:zfos^-1^505d6.3, and zp3e(zgc:171779), that the developmental pathway extended much further in the Liu data than the Hsu data. This isn't a huge concern, but it should be mentioned and discussed if there is a methodological difference that could explain this difference. Maybe it's just the age difference?

Yes, the difference is due to the age difference. Our data was obtained from 26-dpf gonad while Liu’s data was obtained from 40-dpf ovary. The 40-dpf ovary contains more developing oocytes than 26-dpf gonad. Therefore the expression pattern of diplotene markers, *CU467646.3, si:zfos^-1^505d6.3*, and *zp3e(zgc:171779),* were extended much further in 40-dpf dataset than in 26-dpf dataset.

P5 'Zebrafish progenitors are round in shape'. Progenitors of zebrafish oocytes…

The progenitor type of germ cell (including Prog-E, Prog-C and Prog-L), which is generally called gonocyte previously. The germ cells are called oocytes only after meiosis begins. We therefore prefer not to change our text here to avoid confusion.

P5 'et al., 2010). To characterize these three progenitor subpopulations,'I'm not sure which three are referred to here. When adding in GSC, we get four. Please clarify.

The three progenitor subpopulations indicate Prog-E, Prog-C and Prog-L. Thanks for your suggestion. We have revised the text to clarify this point.

'et al., 2010. To characterize the GSC, Prog-E, Prog-C, and Prog-C(S) progenitor subpopulations,'P5 'hybridization (FISH) and detected GSC as positive'. Make plural GSCs.P5 Characterization of the three stages of progenitor subtypes using nanos2, foxl2l, and rec8 was useful.P5 'premature stop codon disrupting the FH domain of Foxl2l'. Spell out 'forkhead'.

Thanks for your suggestion. We have done the revision.

Figures 2C and 2D. These would be more impactful if error bars were displayed on the graphs and numbers of trials or numbers of individuals were shown.

Thanks for your suggestion. We have revised figures 2C and 2D accordingly.

P5 'non-meiotic cystic germ cells were found in foxl2l mutant (Figure 2E).'Plural: mutantsP5 'foxl2l mutant germ cells are arrested at committed progenitor.'. 'foxl2l mutant germ cells are arrested at the committed progenitor stage.'

Thanks for your suggestion. We have done the revision.

P5 '(Supplementary Figure S6A). The expression of the 37 top WT marker genes'. Again in a couple of words, explain the definition of 'top' genes more precisely.

For each cluster, genes that satisfy the following criteria: (i) specificity ≥ 0.2, (ii) Q-value < 0.01, and (iii) having non-zero counts in ≥ 80% of cells in the cluster were selected as markers. Marker genes were ranked based on the specificity within WT clusters, and the top 2 to 4 markers in the rank in each WT cluster were selected as top markers. We have defined the “37 top WT marker genes” more clearly in the paragraph “*Cluster marker identification*” within the section “**Materials and methods**”.

P5 'found that these marker genes were expressed at similar stages as they represented for WT cells.'. The text here needs to say that WT and mutant cells were co-clustered. Also, it would be interesting to see how the pseudotime navigates its way through this blob of cells.

Thank you for your feedback. The mutant analysis was performed exclusively on mutant cells and did not involve co-clustering with WT cells. The UMAPs presented in Figure 3—Figure Supplement 1A (original Figure S6B) are derived solely from mutant cells and underscore this focus. We have clarified this distinction in the revised main text to ensure accuracy and transparency.

Figure 3D. Label the horizontal axis as cluster number. Also, consider changing 'dominant' in the legend to 'predominant'.P6 'integrated groups (Figure 3D). Dominant stages were obtained'. Define' Dominant stages'.

Thanks for your suggestion. The term “dominant” may not be precise. We have now changed it to “prominent” as suggested. The prominent stage of an integrated group was defined as the stage with a significantly higher proportion of cells in the group among the entire WT or mutant population. The “prominent stages” were defined in the section “Composition analysis” within **Materials and methods**.

P6 'calibration against their prevalence rates in the entire cell population.'Does 'entire cell population' mean all of the mutant cells? or all of the mutant plus WT cells?

Thanks for your suggestion. The “entire cell population” indicates either entire WT cells or mutant cells. We have done the revision.

P6 'For mutant cells, while groups I5-I8 had two dominant stages,'If dominant means the one that's present most frequently, then more than one shouldn't be dominant unless there's a tie. So 'dominant' needs a clear definition.

Dominant” might not be accurate when more than one stages were mentioned. We have changed it to “prominent” stages.

P6 'that WT and mutant cells with high cell-to-cell similarity belong to similar developmental stage.''that WT and mutant cells that have high transcriptome similarity likely belong to the same developmental stage.'P6 'Groups I3 and I6 are particularly unique.''Groups I3 and I6 are unique.' Unique means the only one of its kind. So there can't be gradations of being unique.Figure 3E. Should the group at the bottom labeled 'Mutant', be called something like 'Mutant-specific cell types'? Because all of the data in the table refer to mutant cells, right?

Thanks for your suggestion. We have changed the term.

P6 'mutant cells additionally expressed zgc:158463'In addition to what? So mutant cells expressed all the genes of WTs in I3 but in addition zgc:158463 which wasn't expressed in the WT cells in I3? The same question for I6 and tp53. Please clarify.

We have revised the text to clarify the statement.

P6 'Furthermore, the Prog-C marker foxl2l was expressed in mutant cells of I1,'So that means that the mutant mRNA had normal stability? Wasn't degraded by nonsense mediated decay?

Yes, the mutant mRNA had normal stability. It was not degraded by nonsense-mediated decay.

P6 'The aberrant foxl2l expression in some earlier GSC and Prog-E plus the accumulation of foxl2l expressing cells in the mutant indicate that progenitor development is impaired in foxl2l mutants.''in the mutants and the failure to find clusters representing more mature stages indicate that progenitor development is impaired in foxl2l mutants' [could that be added or not?]P7 'throughout the development'. Change to: 'throughout development'P7 'Stratified by integrated groups, we compared'. Check to see if there's a better word than 'stratified' to use here. As written, it means that 'we', the authors, were arranged into layers by integrated groups.

Thanks for your suggestion. We have done the revision.

P7 'WT and mutant cells at the same dominant stages'. Each time that 'dominant' is used, see if 'predominant' would be a better word.

Thanks for your suggestion. To make a more precise statement, we change the term “dominant” into “prominent”

Figure 4A. The triangles are very difficult to see. Would some other visualization work better?

Thank you for the feedback. We have revised Figures 3C and 4A so that they are easier to read.

P7 'Figure 4B shows genes that were differentially expressed between WT and mutants.'Are these all of the genes that were differentially expressed between these two genotypes? Or a subset? And if a subset, then how were these specific genes selected to display?In addition, be explicit about the comparison. Does red mean that the WT or the mutant is overexpressing? E.g., is Dmrt1 overexpressed in WT or in mutant?

Figure 4B shows all differentially expressed genes. The fold changes were calculated by mutant/WT. The red color means the mutant is overexpressing.

Figure 4B. The gradient in the main portion of the figure should match the order shown in the scale. The darkest red indeed is on top, but then for the blues, the darkest is on top, not at the bottom as in the scale.

The genes in the heatmap were ordered by the fold change. In each matched stage, the differentially expressed genes (DEGs) are ordered by the absolute value of the fold change within the set of upregulated or downregulated genes, with the most significantly different genes positioned at the top of each matched stage. This ensures that both positively and negatively regulated genes are prominently displayed. The gradient order reflects this logic, where the darkest red indicates the largest positive fold change and the darkest blue indicates the largest negative fold change. We chose this representation to emphasize the most differentially expressed genes, regardless of the direction of change.

Figure 4c. If id1 and nanos2 are marker genes for GSCs and disappear at later stages in WTs, and if their expression doesn't disappear in mutants, then the mutant cells don't advance to the stage where expression of these genes normally disappear, right? So there wouldn't be any mutant cells at stages G-P to Prog-C(S) because id1 and nanos2 are still expressed in mutant oocytes in these columns in the figure? Or at least these would be variants of the wild type clusters.

*nanos2* is a GSC marker gene while *id1* is not. *Id1* was barely detected in all types of germ cells in WT juvenile gonad. In the mutant, *nanos2* was only expressed in mutant GSC, became barely detected at G-P and Prog-E stages, then was highly expressed at Prog-C stage. The discontinuous expression of *nanos2* in mutant cells during developmental stages suggests that in the mutant, *nanos2* is re-expressed in Prog-C without going through the steps of G-P and Prog-E. We still detected mutant cells that correspond to G-P and Prog-E as shown in Figure 3D.

In addition, it looks like foxl2l inhibits its own expression at early stages, because without foxl2l activity, the expression of foxl2l is much higher in mutants and WTs in GSCI to Prog-E columns.

The *foxl2l* transcripts were aberrantly upregulated in only mutant Prog-E. The expression of *foxl2l* transcripts in mutant GSCI is not significantly different from that in WT, and transcripts in mutant GSCII and P-G stage were really low. It is possible that Foxl2l may inhibit its own expression in Prog-E although we did not investigate this point further.

P7 'The expression of Dmrt1 and dele1 in WT was steady throughout the developmental stages (Figure 4C), but Dmrt1 was abnormally high and dele1 was abnormally low in both mutant GSCI and Prog-C/Prog-C(S).'The fact that Dmrt1 was abnormally high in mutants suggests that a role of foxl2l activity is to inhibit expression of Dmrt1.The fact that dele1 was low in mutant Prog-C(S) cells in mutants, suggests that foxl2l activity encourages expression of dele1.Those conclusions, if authors agree with them, might have more impact here than what's written: 'These results indicate multiple defects of gene expression in the mutants.'P7 'Co-expression of foxl2l with Dmrt1, nanos2 or id1 in foxl2l mutant'. Consider changing to '…in foxl2l mutants', or to '…in foxl2l mutant germ cells'.Figure 5A legend. Say explicitly that cells expressing both genes are represented by yellow. It's obvious of course from the scale grid but would help some readers to say it explicitly.P7 'increased in the mutant (Figure 5B). Moreover, the number of nanos2+ cell'Plural: …of nanos2+ cells…'P7 '(Figure 5C). To further investigate whether GSC is affected in mutants' '…whether GSCs are affected…'

Thanks for your suggestion. We have done the revision.

P7 'counted the number of nanos2+foxl2l- cells, and found that the number increased dramatically in mutants at 26 dpf'. increased dramatically in mutants compared to WTs? Or increased in mutants at 26 dpf compared to some other time in development?

Thanks for your suggestion, we have done the revision (compared with WT).

P7 'GSC cells in mutant together'. Plural: 'GSC cells in mutants together'

Thanks for your suggestion, we have done the revision.

P7 'GSC cells in mutant together suggest that some Prog-C may revert to GSC in', What is the evidence that they develop to Prog-C and revert rather than that they never made it fully to Prog-C and simply maintained some of the gene expression properties of the earlier stages? These are two competing hypotheses, and both should be mentioned.

We have examined the expression profiles of GSC markers including *nanos2*, *cbx7a*, *psmb13a* and *dnmt3bb.1* in mutant cells. We found that the expression of these genes was low in mutants at GSC-to-progenitor transition stage and Prog-E stage. Their expression remained low after cells are differentiated into progenitors, and only *nanos2* was abruptly upregulated in mutant Prog-C. Therefore, it is more likely that the mutant Prog-C acquires some GSC property with the upregulation of *nanos2* instead of the maintenance of GSC property during the development from GSC to progenitor. Nevertheless, we followed the suggestion of this reviewer and mentioned both hypotheses in this section.

Figure 6 legend. Mention 'G-P transition' in the legend or remove it from the sketch. The word 'Proliferation' in black on a dark gray field is hard to read.

Thanks for your suggestion. We have revised the figure and the figure legend.

P9 'they undergo G to P transition for further differentiation into progenitors.'Define 'G-to-P'.

Thanks for your suggestion. We have revised the sentence.

P8 'Progenitors include three types: Prog-E, Prog-C, and Prog-L.' It would be good if the text here could state the transcriptomic features that distinguish these three phases from each other like the text did for the two kinds of GSCs.

Thanks for your suggestion. We have done the revision to include transcriptomic features of these cell types.

P9 'The fate of germ cells is determined following the instruction of the supporting cells.' Compare to germ cell sex determination in mammals.

Thanks for your suggestion, we have done revision to describe the germ cell sex determination in mammals.

P9 'deficiency, however, leads to cell arrest at Prog-C as shown in this article.'. 'deficiency, however, leads to cell arrest at Prog-C as shown by results presented here.

Thanks for your suggestion, we have done the revision

P9 'commitment to the female fate as shown here. It indicates'. What is the antecedent of 'It'?

We have rearranged this paragraph to avoid ambiguity in our statement.

P10 'Id1, inhibitor of DNA binding 1, is a transcription regulator that lacks DNA…'.'…that lacks a DNA …'

Thanks for your suggestion, we have done the revision.

P10 'stem-cell genes (id1 and nanos2) and male gene Dmrt1.' First, '…and the male gene Dmrt1'Second, are there other early male genes that are suppressed by Foxl2l action besides Dmrt1?

Thanks for your suggestion. Among the total 43 DEGs shown in Figure 4B, *Dmrt1* is the only gene known for male differentiation in zebrafish and mammals.

P10 'The alternative development of foxl2l mutant germ cell'. Plural: '…mutant germ cells'

Thanks for your suggestion, we have done the revision.